# Dynamical latent state computation in the male macaque posterior parietal cortex

Kaushik J. Lakshminarasimhan [1] ✉, Eric Avila[2], Xaq Pitkow [3,4,5,7] &
Dora E. Angelaki[2,6,7]

Success in many real-world tasks depends on our ability to dynamically track hidden states of the world. We hypothesized that neural populations estimate these states by processing sensory history through recurrent interactions which reflect the internal model of the world. To test this, we recorded brain activity in posterior parietal cortex (PPC) of monkeys navigating by optic flow to a hidden target location within a virtual environment, without explicit position cues. In addition to sequential neural dynamics and strong inter-neuronal interactions, we found that the hidden state - monkey's displacement from the goal - was encoded in single neurons, and could be dynamically decoded from population activity. The decoded estimates predicted navigation performance on individual trials. Task manipulations that perturbed the world model induced substantial changes in neural interactions, and modified the neural representation of the hidden state, while representations of sensory and motor variables remained stable. The findings were recapitulated by a task-optimized recurrent neural network model, suggesting that task demands shape the neural interactions in PPC, leading them to embody a world model that consolidates information and tracks task-relevant hidden states.

Imagine you are driving on a busy highway and wish to change lanes. To safely do so, you need to mentally track the pattern of traffic behind you even when not looking into the rear-view mirror. Many everyday tasks require maintaining and updating beliefs about state variables that are not directly observable. This can be computationally hard especially if the latent world states are continuous-valued, i.e., assume a range of values, and dynamic (vary in time); these properties are typically true in the real-world[1]. Mechanisms underlying sensory perception and movement generation have been extensively investigated under a wide variety of conditions, such that we are converging on good computational models that are consistent with neural data[2–5]. In contrast, we do not understand how the intermediate, continuous-valued, time-varying, latent states - the stuff of thoughts - are represented in the brain, nor the mechanisms used to compute those states[6,7]. Filling this void is

essential to building a complete picture of neural computations in the sensorimotor loop.

The past few decades have seen the emergence of two distinct approaches in the study of neural representation of latent world states. These have contributed significantly to our understanding in complementary ways. One approach, following the tradition of sensory neuroscience, uses binary decision-making tasks (e.g., motion direction discrimination) in which participants gradually integrate sensory evidence over time and then report one perceived outcome (e.g., dots moving to the left or right)[8–11]. The high degree of experimental control afforded by this paradigm has helped reveal a tight link between the neural activity in the posterior parietal cortex and the time course of decision variables that guide behavior[12,13]. However, because the latent world states themselves tend to be discrete and/or static in such tasks, it is difficult to fully extrapolate those insights to continuous,

[1]Center for Theoretical Neuroscience, Columbia University, New York City, NY, USA. [2]Center for Neural Science, New York University, New York City, NY, USA. [3]Department of Neuroscience, Baylor College of Medicine, Houston, TX, USA. [4]Center for Neuroscience and Artificial Intelligence, Baylor College of Medicine, Houston, TX, USA. [5]Electrical & Computer Engineering, Rice University, Houston, TX, USA. [6]Department of Mechanical and Aerospace Engineering, New York University, New York City, NY, USA. [7]These authors jointly supervised this work: Xaq Pitkow, Dora E. Angelaki. ✉e-mail: jl5649@columbia.edu

interactive behaviors where those latent states change continually as a consequence of one's own actions. The alternative approach, which emerged from cognitive psychology, has sought to characterize neural correlates of continuously changing latent world states (e.g., position, heading of freely foraging animals)[14,15]. This has led to a rich description of neural maps in the hippocampal formation that can potentially be used for computing latent world states. However, because neither sensory input nor behavior is controlled in this approach, it is difficult to determine the precise relationship between neural activity and the animal's momentary beliefs in such settings. To overcome these limitations, we took an approach that combined the desirable elements of both approaches by using a task that was ecologically valid, yet well-defined and controllable. Our goal was threefold: (i) to characterize the neural representation of the repertoire of sensory, latent, and motor variables in a naturalistic closed-loop task featuring action-perception loops, (ii) to test whether the latent states computed by the neural population influence behavior, and (iii) to constrain the space of possible mechanisms that create the neural representation of the latent states.

We created a virtual environment in which monkeys used a joystick to steer to a transiently cued, random target location by integrating sparse optic flow cues[16]. To successfully perform the task, monkeys had to continuously update an internal estimate of the relative target location (the latent state) by integrating their own movement velocity inferred from the sparse optic flow cues. Brain regions in the posterior parietal cortex (PPC) have been implicated in various aspects of this computation such as optic flow processing[17,18], working memory[19–21], as well as planning of spatial movements[22,23]. Because we are primarily interested in understanding the mechanisms of latent state computation rather than optic flow processing per se, we wanted to record neural activity in a region within PPC that likely already receives abstract velocity signals, such that it may serve as the locus of latent state computation in our task. There are several properties that make area 7a a more ideal candidate than other parts of PPC. First, anatomical tracing studies have consistently found a pattern of interareal connectivity that places area 7a at the top of the motion-processing ('dorsal stream') hierarchy[24,25]. Moreover, it is one of the few areas in PPC that directly projects to the hippocampal formation[26,27], with lesions to area 7a affecting navigation performance[28,29]. Second, area 7a neurons are known to have large, bilateral receptive fields (15–25 degrees) and activated by the full-field motion stimuli used in our VR environment[30]. Third, response properties of area 7a neurons indicate that they are capable of marginalizing away the influence of eye movements thereby representing visual inputs in a navigationally useful, non-retinotopic format at the population level[31]. Fourth, we confirmed in prior work under passive viewing conditions that neurons in area 7a indeed encode linear and angular velocity in an abstract format, regardless of stimulus modality[18]. Finally, previous work has shown that representation of cognitive variables in area 7a is clearly decoupled from the influence of sensory and motor variables[32,33] whereas such decoupling has not been demonstrated elsewhere in PPC. Therefore, we simultaneously recorded from a large number of neurons from area 7a of PPC while monkeys performed this task.

We found that neural populations exhibited sequential activity during this task, and that coupling between neurons contributed substantially to the neural activity. Furthermore, single neurons carried information about sensory, latent, and motor variables, and latent world states decoded from the population activity were predictive of monkeys' behavioral errors on individual trials. Finally, task manipulations that perturbed the world model dramatically altered both neuronal coupling and latent state tuning, but only minimally affected tuning to sensory and motor variables. These results suggest that PPC maintains dynamical beliefs about latent world states during naturalistic behaviors involving action/perception loops.

## Results

Three monkeys performed a visual navigation task in which they used a joystick to steer to a transiently cued target location in a three-dimensional virtual reality (VR) environment without allocentric reference cues (i.e., stable landmarks) (Figs. 1a and S1a and "Methods"). Individual visual elements comprising the ground plane were visible only transiently and could not be used as landmarks. At the beginning of each trial, a circular target on the ground plane blinked briefly at a random location within the field of view, and then disappeared. The joystick controlled forward and angular velocities, allowing subjects to steer freely in two dimensions (Fig. 1b–left). The goal was to steer toward the target and stop when their position fell within a circular reward zone centered on the target (Fig. 1b–middle). The joystick was controlled via a mixture of frontal and lateral hand movements (Fig. 1b–right and Fig. S1b). On each trial, a target location was drawn randomly from a uniformly distribution over the ground plane area within the subject's field of view (1–4 m, ±40°; Fig. 1c–left), eliciting diverse steering maneuvers as seen from their movement trajectories across trials (Fig. 1c–right). Performance feedback was provided at the end of each trial in the form of juice reward for correctly stopping within the reward zone (0.6 m radius; Fig. 1d) after waiting for a variable delay period (0.2–0.6 s).

### Behavioral performance

Because target locations were randomized, travel durations varied widely across trials (median ± interquartile range [IQR]: $1.9 \pm 0.8$ s). On average, $61.5 \pm 4\%$ of the trials were rewarded, and the average error in stopping position was 0.41 ± 0.1 m. Both radial distance (Fig. 1e–left) and angular eccentricity (Fig. 1e–right) of the monkeys' responses (stopping location) were highly correlated with the target location across trials (Pearson's $r \pm$ SD, radial: 0.71 ± 0.06, angular: 0.87 ± 0.05). To test whether performance was accurate, we regressed responses against target locations. The slope of the regression was close to unity both for radial distance (0.87 ± 0.04) and angle (0.94 ± 0.06), suggesting that the monkeys were nearly unbiased. Non-parametric regression yielded qualitatively similar results (Fig. S1c), but additionally revealed modest undershooting for the most distant targets, an effect that is likely due to growing position uncertainty described in previous work[34].

Although the above results suggest that the behavior was appropriately modulated by task demands, they do not satisfactorily capture the performance for two reasons. First, they ignore differences in task difficulty associated with varying target distance. Second, they do not account for the errors arising from intrinsic variability in motor commands. Therefore, we used an approach that is conceptually similar to receiver operating characteristic (ROC) analysis to objectively evaluate the performance by accounting for both sources of variability. For each behavioral session, we constructed a "psychometric function" by computing reward probability as a function of a hypothetical reward window size (Fig. 1f–left; "Methods"). By plotting the true psychometric function against one obtained by shuffling target locations across trials, we obtain the monkey's ROC curve (Fig. 1f–right). Chance-level performance would correspond to an area under the ROC curve (AUC) of 0.5, while perfectly accurate responses (zero error) will yield an AUC of one. The AUCs were well above chance (mean ± SD, 0.88 ± 0.03; Fig. 1f–right inset) and stable across target distances and angles (Fig. S1d). Nonetheless, the AUC was significantly worse than a subset (10%) of interleaved trials in which the target was visible throughout (0.94 ± 0.04, $p = 0.002$, two-sample $t$ test; Figs. 1g and S1e). This suggests that monkeys found this task quite challenging, and performance was not limited simply by motor variability.

In principle, it is possible to avoid integrating optic flow by learning the precise transformation implemented by the joystick controller. However, as we will show later and as described in previous work, monkeys are sensitive to multiple task manipulations in a

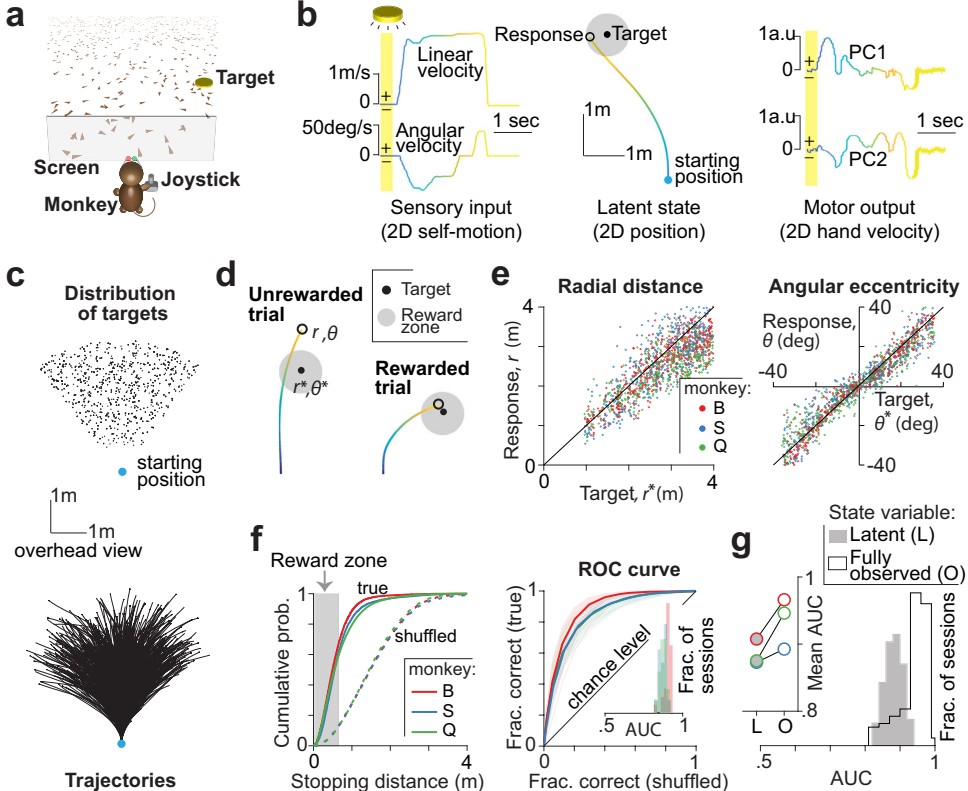

**Fig. 1 | Monkeys navigated to a remembered goal by integrating optic flow.**
**a** Monkeys used a joystick with two degrees-of-freedom to navigate to a cued target (yellow disc) using optic flow cues generated by ground plane elements (brown triangles) in a virtual environment. **b** Left: The time course of sensory input variables—linear (top) and angular (bottom) velocities—during one example trial. Middle: Overhead view of the spatial position of the monkey during the trial. Black open circle denotes the monkey's response (stopping location). As there are no visual landmarks, the position becomes latent to the monkey as soon as the target is turned off. Right: Monkey's hand velocity along two leading principal components of hand position, while maneuvering the joystick. Yellow shaded regions correspond to the time period (-300 ms) when the target was visible on the screen. Time is also coded by color. **c** Top: Overhead view of the spatial distribution of target positions across trials. Bottom: Movement trajectories of one monkey during a representative subset of trials. Blue dot denotes starting location. **d** Example trials showing "incorrect" (left) and "correct" (right) responses of a monkey. **e** Left: Comparison of the radial distance of the response against radial distance of the target across a subset of trials from three different monkeys. Right: Angular eccentricity of the response versus target angle. Black diagonal lines have unity slope. The starting position was taken as the origin. **f** Left: Cumulative distribution of stopping distance (from the target center) across trials of the three monkeys. Dashed curves show the corresponding null distribution calculated by shuffling response and target locations. Gray region highlights the range of stopping distances that guaranteed reward. The cumulative probability of an arbitrary stopping distance can also be interpreted as the hit rate (fraction of correct trials) if that stopping distance was taken to be the edge of the reward zone. With this interpretation, we can construct ROC curves by plotting the true hit rates against shuffled hit rates across the range of stopping distances. Right: ROC curves from the three monkeys, averaged across sessions. Data from individual recording sessions are overlaid in thin lines. Inset—Histograms of the area under the corresponding ROC curves (AUC). **g** In a random subset (10%) of the trials, the target remained visible throughout such that the world state was fully observable to the monkeys. Histograms of the AUCs for trials in which the world state was latent (gray shaded) or fully observable (black open). Trials were pooled across monkeys. Inset —Mean AUCs of individual monkeys under the two conditions (L latent, O fully observable). **a**–**c** reprinted from Lakshminarasimhan et al.[16], Copyright (2020), with permission from Elsevier. Source data are provided as a Source data file.

manner that strongly suggests that they perform the task by integrating optic flow to update beliefs about their spatial position[35]. In the following sections, we examine the neural dynamics during this task, including the neural representation of dynamically evolving latent state estimates about position and other task variables.

## Neural dynamics

We recorded neural activity from the PPC (area 7a) using chronically implanted multi-electrode arrays, while monkeys performed the task (Methods, Fig. 2a). A total of 1612 units were recorded across 32 sessions ($44 \pm 12$ units/session). To avoid double counting neurons, we restrict our focus to a subset of 244 neurons obtained from three sessions with the highest yield, one from each monkey. Data from the remaining sessions are analyzed and presented in Supplementary Material for comparison. Because this task challenged monkeys to integrate self-motion and update beliefs about their position relative to a remembered target throughout the trial, it places a significant strain on working memory. Classic working memory paradigms have

found either persistent activity of single neurons or activation of many neurons in sequence. Few neurons in our data exhibited persistent activity during the trial. Instead, neurons seemed to be more active at certain periods of the trial, with some neurons being active earlier than others (Fig. 2b—compare #1, #2 vs #3, #4). Therefore, we wanted to test if neurons instead exhibited sequential activity dynamics at the population level.

Since changes to the latent state are restricted to the time between target onset and the end of movement, we estimated population firing rate maps by rescaling time over this period and computing the trial-averaged response of each neuron. We sorted the neurons according to the timing of peak activity ("Methods"). We found strong sequential activation of neurons in all three monkeys, as quantified by a standard index of sequentiality (Sql) that ranges from 0 (random) to 1 (sequential) ("Methods," Mean Sql $\pm$ 95% CI—Monkey B: $0.34 \pm 0.1$, Monkey S: $0.23 \pm 0.12$, Monkey Q: $0.28 \pm 0.1$). Furthermore, the degree of sequentiality was robust to task demands: Sql was similar across groups of trials corresponding to different target distances

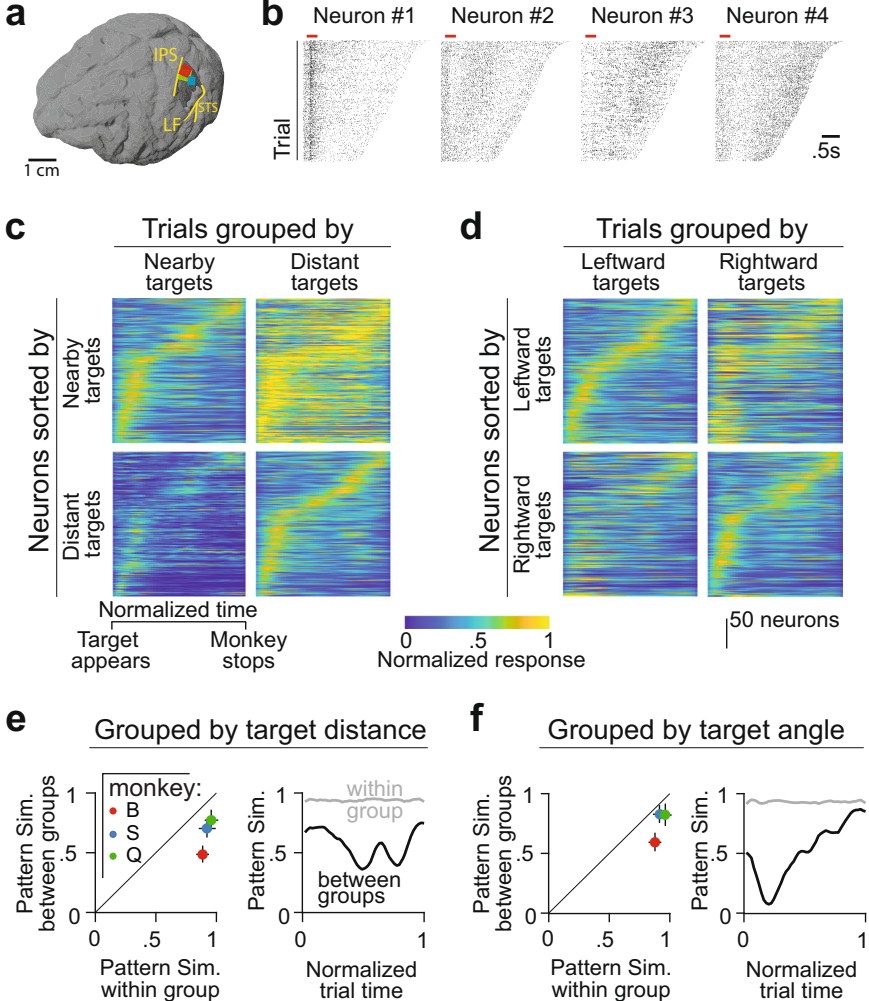

**Fig. 2 | Latent states dictate population dynamics. a** Anatomical location of the multi-electrode arrays (Red−monkey B, Green−monkey Q, Blue−monkey S) superimposed on the 3D reconstructed brain of monkey S (IPS−intraparietal sulcus, STS−superior temporal sulcus, LF−lateral fissure). **b** Spike rasters of four example neurons from one of the recording sessions. Red bar denotes the 300-ms period during which the target appeared on the screen, and rasters are shown until the end of the stopping period while monkeys waited for feedback. **c** Peak-normalized response of neurons calculated by averaging across the set of trials with nearby (left panels) or distant (right panels) targets. Neurons are sorted according to the timing of their peak response observed in the set of trials with nearby (top panels) or distant (bottom panels) targets. Spike times were rescaled based on the trial duration before trial-averaging and the resulting response profile of each neuron was subsequently normalized by the peak activity observed in the condition used for sorting. Neurons from all three monkeys are combined before sorting (see Fig. S2a, b for individual monkeys). **d** Similar to **a**, but with trials grouped by target angle. **e** Left: Comparison of the pattern similarity of the population dynamics between trials within the same (abscissa) or different (ordinate) groups, shown separately for each monkey. Pattern similarity was defined as the correlation coefficient between the firing rate maps taken from either the same trial group (odd vs even trials) or different trial groups (nearby vs distant targets). Right: Time course of the pattern similarity, computed as the correlation between population activity vectors (columns of the rate maps) taken from the same trial group (odd vs even trials) or different trial groups (nearby vs distant targets). **f** Similar to **c**, but with trials grouped by target angle. In **e**, **f**, $n = 112$, $n = 68$, $n = 64$ neurons in monkey B, S and Q, respectively. Error bars denote ±1 SEM. Source data are provided as a Source data file.

(Mean Sql ± 95% CI, Nearby targets: 0.21 ± 0.1, Distant targets: 0.19 ± 0.08) and target angles (Leftward targets: 0.26 ± 0.07, Rightward targets: 0.22 ± 0.11). However, the overall activity level and the precise sequence in which neurons were activated was not preserved across trial groups: normalizing the activity and sorting the neurons according to one group of trials did not yield identical sequential activity in the complementary trial group (Figs. 2c, d and S2a, b). To quantify this, we computed the pattern similarity of population rate maps between task-relevant trial groups, and found that it was significantly lower than the similarity between odd and even trials within each trial group (Pattern similarity, Odd vs Even trials: 0.92 ± 0.03, Nearby vs Distant: 0.65 ± 0.04, Leftward vs Rightward: 0.72 ± 0.03; Fig. 2e, f−left). This suggests that the variability in sequencing is not due to neural noise, but rather reflects systematic representation differences between different trial groups. The pattern similarity was low throughout the trial and not just at the beginning or the end of the trial (Fig. 2e, f−right, Fig. S2c, d), suggesting that task demands alter the population dynamics and neurons are not merely keeping time.

While sequential neural dynamics could partly be a signature of the temporal integration process by which monkeys update their position estimates, sensory cues (optic flow) and motor commands (hand motion) are trajectory-dependent and thus also differ across trial groups. Consequently, to test whether PPC contains all signals relevant to this task, we need an explicit encoding model to relate neural activity to dynamical latent state estimates as well as other variables that influence neural activity, such as sensory and motor variables. Furthermore, to the extent that the activity is not driven solely by external inputs, the presence of sequential dynamics points to a potentially important role for neural interactions within PPC. We next fit a model with these features.

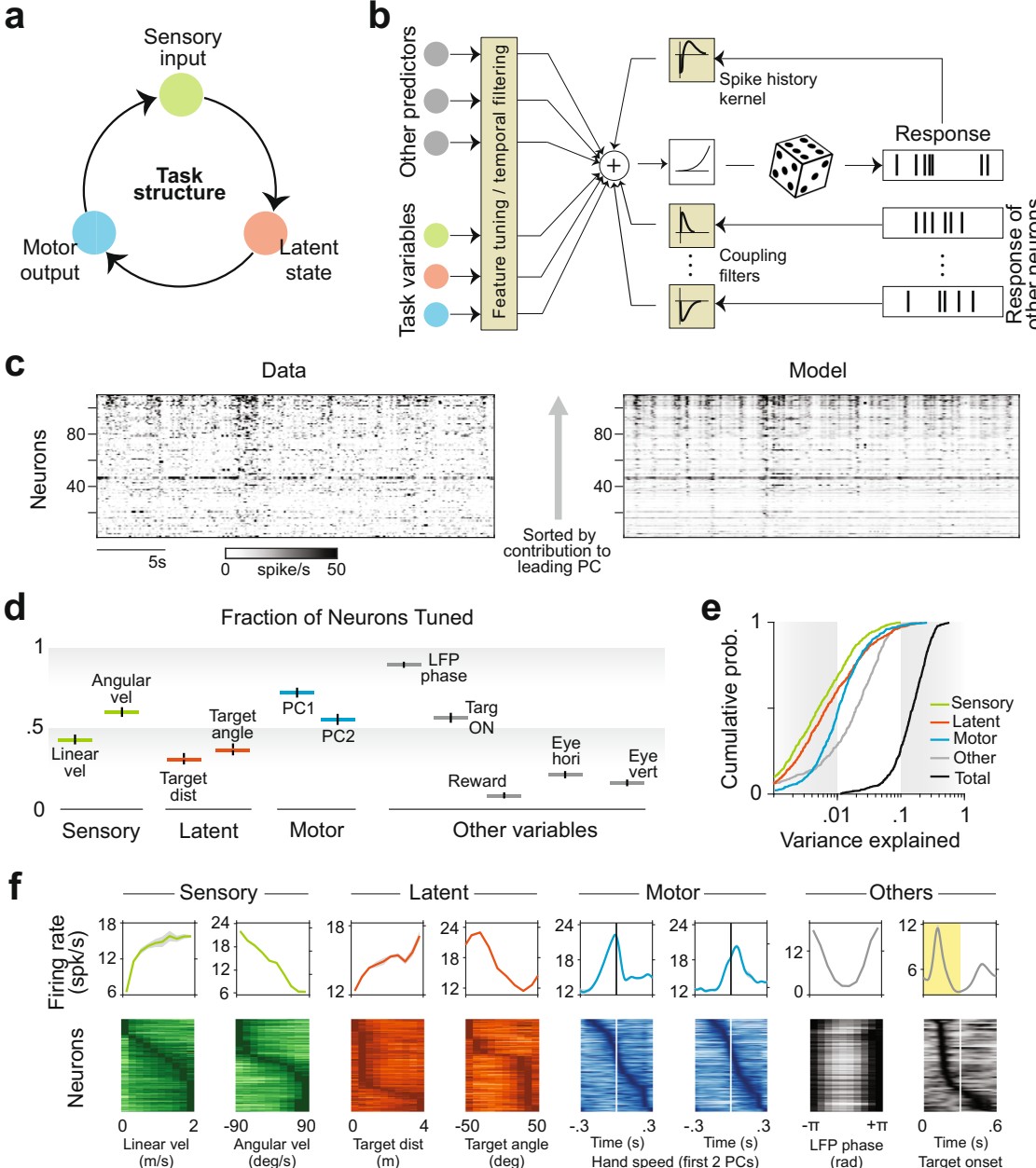

**Fig. 3 | Neurons represent sensory variables, latent world states, and motor variables. a** Causal structure of the task, illustrating the recurrent nature of the interaction between sensory inputs, latent states, and motor outputs. **b** Schematic of the generalized additive model (GAM) used to fit spike trains of single neurons. **c** Activity of simultaneously recorded neurons during a random thirty second epoch during the experiment (left) and the corresponding prediction reconstructed using the model (right). Neurons are arranged according to their contribution to the leading principal component (PC) (bottom−lowest; top−highest) for visualization. **d** Proportion of neurons tuned to different variables. Error bars denote ±1 standard error of binomial proportions (*n* = 244). **e** Cumulative distribution of the contribution of different predictors, calculated as the reduction in variance explained by the model after removing those predictors. Black shows the distribution of the variance explained by the full model. **f** Top: Example tuning functions showing sensitivity of neurons to different task variables and other explanatory variables. Shaded regions denote ±1 SEM across validation sets (*n* = 10). Bottom: Peak-normalized tuning functions of all significantly tuned neurons, sorted according to the peak feature. Across the population, tuning to individual task variables had peak responses that tiled all values of the feature space. In contrast, most neurons produced stereotyped responses to Local field potential (LFP) phase. Source data are provided as a Source data file.

## Encoding model

Because the causal structure of the task involves an action-perception loop, all task variables change dynamically during the course of the trial (Fig. 3a). To simultaneously account for how neural activity is influenced by dynamic sensory (linear and angular velocity), latent (target distance and angle), and motor predictors (hand speed along the two principal components of hand position) in addition to the monkey's gaze and discrete events (target onset and reward delivery), we fit a generalized additive model (GAM) with Poisson distributed spike counts ("Methods"; Fig. 3b). Consistent with previous work in the monkey PPC, we observed robust ~15Hz oscillations in the local field potential (LFP, Fig. S3a). Therefore, we included the LFP phase as a predictor to capture temporal structure in the spike trains associated with these global rhythms. Finally, the model also incorporated temporal filters that explicitly captured causal, directional functional coupling between neurons, and autoregressive effects (spike-history filter). The model differs from a traditional encoding model in that it fits arbitrary nonlinear mappings (tuning functions) from predictors to

neuronal response rather than linear kernels. However, this approach is closely related to generalized linear models (GLMs) in neuroscience[36], and is in fact identical to it if the task predictors are first expressed in terms of an appropriate non-linear basis. An iterative pruning procedure (backward elimination) identified task variables to which individual neurons were significantly tuned.

The best-fit GAM captured $58 \pm 6\%$ of variance in the structure of population response (Fig. 3c) and $15 \pm 4\%$ of the temporal variability in single neuron responses (Fig. S3b) suggesting that the model had good predictive power. A significant fraction of neurons was tuned to linear and angular velocity ($52 \pm 8\%$), target distance and angle ($37 \pm 5\%$) and motor output ($58 \pm 4\%$) (Fig. 3d). The majority of neurons ($53 \pm 4\%$) were driven by target presentation while few neurons showed sensitivity to reward delivery ($7 \pm 4\%$). Because different task signals were mixed at the level of single neurons (Fig. S3c), the variance explained by individual task variables was typically low (Figs. 3e and S3d). Examining the model parameters, we found that single neurons were often tuned to sensory and motor variables, as well as to latent world states i.e., target distance and angle (Figs. 3f and S3e). Across the population of neurons, we found near uniform tiling of the sensory space with a significant fraction of neurons tuned for low, intermediate, and high linear and angular velocities (Fig. 3f—green). A similar trend was seen in the temporal kernels fit to motor variables (hand speed) where we found a good mix of neurons that responded before and after hand motions (Fig. 3f—blue). On the other hand, tuning to target distance and angle were relatively skewed, with the majority of neurons tuned to small target distances and extreme target angles (Fig. 3f—orange). Greater preference for small target distances could be a reflection of the fact that those states carry the highest value in the context of the task. Tuning to LFP phase was highly stereotyped: the spiking probability of all neurons peaked just before ($18 \pm 8°$) the trough of the LFP signal. Tuning to target onset was somewhat variable, with the vast majority of neurons (71%) exhibiting an ON response with a latency of $122 \pm 15$ ms, and a smaller group of neurons (14%) which responded exclusively after the target turned OFF with a latency of $64 \pm 32$ ms (Fig. S3f). Tuning to gaze position was also diverse (Fig. S3g), consistent with the diversity of 'gain fields' discovered by classic studies investigating the role of posterior parietal cortex in transforming visual input from retinal co-ordinates into actionable co-ordinates[31].

Next, we tested the extent to which coupling and spike-history filters contributed to neuronal responses. To do this, we compared likelihoods of the model with neither coupling nor spike-history filters against the model that included them both (Fig. 4a). We found that the likelihood was substantially greater for the coupled model, and marginally better for the model with just the spike-history factor (Mean Likelihoods $\pm$ SD across neurons, Uncoupled model: $0.36 \pm 0.1$, Spike-history model: $0.39 \pm 0.1$, Coupled model: $0.65 \pm 0.14$). The amplitude of these filters capture the modulation ('gain') in the probability of spiking as a function of the time since last spike from either the same neuron or another neuron (Fig. 4b—top). By doing so, they are able to capture features of spiking that are distinct from features captured by the task variables. In particular, the spike-history filter and coupling filter capture autocovariance and cross-covariance between neurons respectively that are not due to fluctuations in task variables (Fig. 4b—bottom). As a result, the coupled model is able to recapitulate the spatiotemporal covariance structure of the population activity very well (Pearson's $r \pm$ SD, Data vs Coupled model prediction: $0.86 \pm 0.08$, Data vs Uncoupled: $0.06 \pm 0.1$; Figs. 4c and S4a), yielding substantially better predictions than the uncoupled model. Note that the uncoupled model can still capture covariance between neurons induced by the fluctuations in task variables, but not shared fluctuations at the millisecond timescale. Moreover, due to the directed nature of coupling filters, the coupled model can capture millisecond timescale, asymmetric interactions between neurons that may arise from recurrent connectivity. We found that the structure of coupling filters was sparse yet diverse: the same neuron had both net excitatory and inhibitory effects on different target neurons (Figs. 4d and S4b). This diversity should not be mistaken for a violation of Dale's law since coupling filters capture effective interaction between neurons, rather than synaptic transmission properties. Across the population of all neuronal pairs, the mean gain was $1.04 \pm 0.05$ suggesting that excitatory and inhibitory effects were balanced (Fig. 4e—top). The timescale of coupling followed a power-law decay, with fast timescales contributing substantially more power to the coupling filter (Fig. 4e—bottom, Fig. S4c). The gain of both excitatory and inhibitory couplings decreased with distance between neurons (Fig. 4f), mirroring widely documented trends in anatomical connectivity and correlated variability.

## Population decoding

We have seen that single neurons in the *macaque* PPC encode task-relevant variables, in particular, the latent i.e., spatial position of the monkey. Because successful performance in this task depends on tracking the dynamical latent state, the above finding indicates that PPC might be critically involved in implementing the underlying sensorimotor transformation. If this is the case, then we can make the following predictions. First, we should be able to dynamically decode sensory, latent, and motor variables with good precision from the population activity. Second, trial-by-trial fluctuations in the error in decoding the latent state from neural activity should propagate to the motor plan and thus should be correlated with behavioral error (latent → motor). Likewise, trial-by-trial fluctuations in the error in decoding the sensory input should be correlated with the error in decoding the latent state (sensory → latent) that depends on those sensory inputs. We tested all three predictions.

To test whether neural activity was informative about task variables, we trained a linear decoder of the population response separately for each task variable, regressing each variable against the activity of all simultaneously recorded neurons ("Methods"). Figure 5a shows the timecourse of different task variables estimated using the corresponding decoding weights, on six example trials. The estimates were remarkably well aligned with the ground truth (gray). The correlation between the true and the decoded values was high, demonstrating good decoder performance (Pearson's $r \pm$ SD, Sensory: $0.77 \pm 0.04$, Latent: $0.66 \pm 0.06$, Motor: $0.74 \pm 0.04$; Fig. 5b). While decoder performance initially increased with population size, the performance began to level off suggesting that the assessment of the decoders was not dramatically limited by the recording size (Fig. S5a).

Next, we assessed whether the decoding performance was correlated with behavior. Because the monkey's decision to stop moving ultimately depended on their (internal) estimate of target distance, we restricted our focus on the decoder of this particular latent variable for this analysis. Both raw estimates of the decoder performance and estimates extrapolated to infinite population predicted monkeys' behavioral accuracy, taken to be the fraction of rewarded trials (Figs. 5c and S5b). As a stronger test, we evaluated the correlation between the decoding error and behavioral error across trials for each monkey. Due to the fine-grained nature of this analysis, we took behavioral error to be the stopping distance to target rather than a binary variable (rewarded/unrewarded). The correlation was significantly greater than chance in all three monkeys (Fig. 5d—left, Fig. S5c). Monkeys tended to undershoot when the decoder underestimated the target distance, and overshoot when there was overestimation (Median decoding errors, Undershot trials: $-13 \pm 3$ cm, Overshot trials: $11 \pm 2$ cm; Fig. 5d—middle). Consequently, we were able to successfully classify undershot/overshot trials with $69 \pm 3\%$ accuracy based on an ROC analysis on the distribution of decoding errors (Fig. 5d—right).

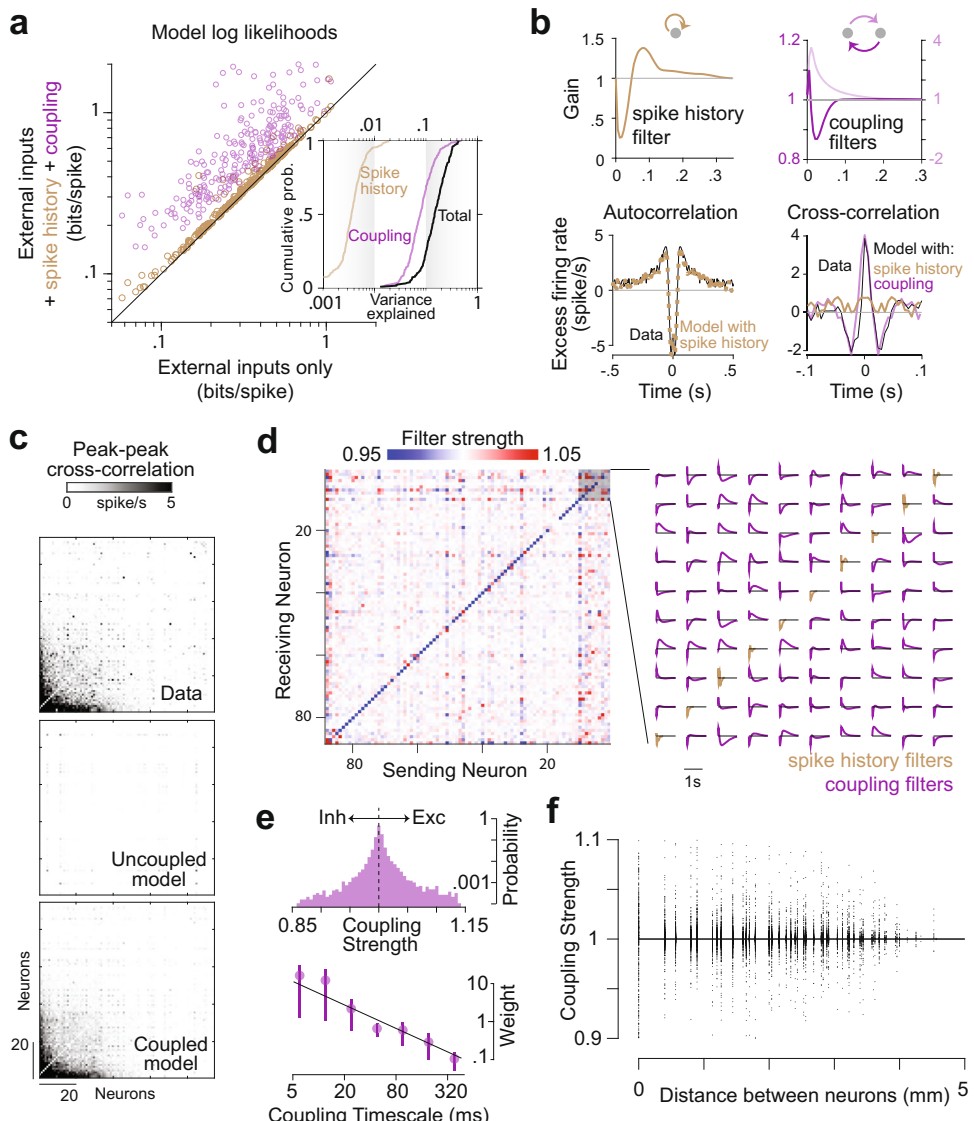

**Fig. 4 | Coupling encapsulates population structure. a** Comparison of the log likelihoods of the model constructed using only external inputs as predictors against those of a model that also included effects of spike-history (brown), against a model with both spike-history and inter-neuronal coupling (maroon). Each circle denotes an individual neuron. Inset shows the cumulative distribution of the contribution of spike history and neuron-neuron coupling, calculated as the reduction in variance explained by the model after removing those filters. Black curve shows the distribution of the variance explained by the full model. **b** Top left: Spike history filter of an example neuron. Bottom left: Model with spike history filter accurately captures the autocorrelation function of the neuron. Top right: Bidirectional coupling filters between an example pair of neurons. Bottom right: Coupling filters capture the cross-correlation between the pair, whereas spike-history filters alone do not. **c** Top: Structure of the peak-to-trough amplitude of cross-correlation between the activity of all pairs of simultaneously recorded neurons from an example monkey. Neurons are ordered according to the weight of their contribution to the first principal component of the population activity. Coupled model (bottom), but not the uncoupled model (middle), captures the structure of cross-correlation of the full population. **d** Left: The strength of the coupling filters

between all pairs of neurons in the population shown in **c**. Strength of the filter was computed by taking the total area under the filter. A strength greater than one corresponds to excitatory coupling whereas less than one corresponds to inhibitory coupling. The diagonal elements correspond to the strength of the spike-history filter (self-coupling) as a special case. Right: Details of the coupling (off-diagonal) and spike-history filters of a subset of the neural population, highlighting the diversity in the filter profiles across neuronal pairs. **e** Top: Frequency distribution over the coupling strengths between all pairs of neurons, pooled across monkeys. A vast majority of the neurons were weakly coupled (note the log scale of the frequency axis). Bottom: Each coupling filter was expressed as a weighted sum of seven exponential basis functions with different decay constants ([6, 12, 24, 48, 96, 192, 384] milliseconds). Data points show the average magnitude of weighting of the different basis functions across all coupling filters, pooled across monkeys. Black line denotes the best fit power-law relationship. Error bars denote ± 1 SEM ($n = 244$). **f** Strength of coupling decreased as a function of distance between the electrodes from which the neurons were recorded. Points above and below the black line correspond to excitatory and inhibitory couplings respectively. Source data are provided as a Source data file.

Finally, we tested whether error in decoding sensory inputs propagates to the latent state representation. Across trials, we found that sensory decoding error was significantly correlated with the error in decoding latent states (Pearson's $r$, Linear velocity vs Target distance: $0.15 \pm 0.1$, Angular velocity vs Target angle: $0.26 \pm 0.1$; Fig. 5e—left and middle, Fig. S5d). Although the weights of sensory and latent decoders

were very different, they were not perfectly orthogonal (Fig. S5e). We controlled for this by computing a null distribution of correlations, by projecting population activity onto pairs of random surrogate modes that were separated by the same angle as the decoders. Correlation between surrogate responses was significantly less that the correlation between error in sensory and latent decoders ($p = 0.008$, paired $t$ test;

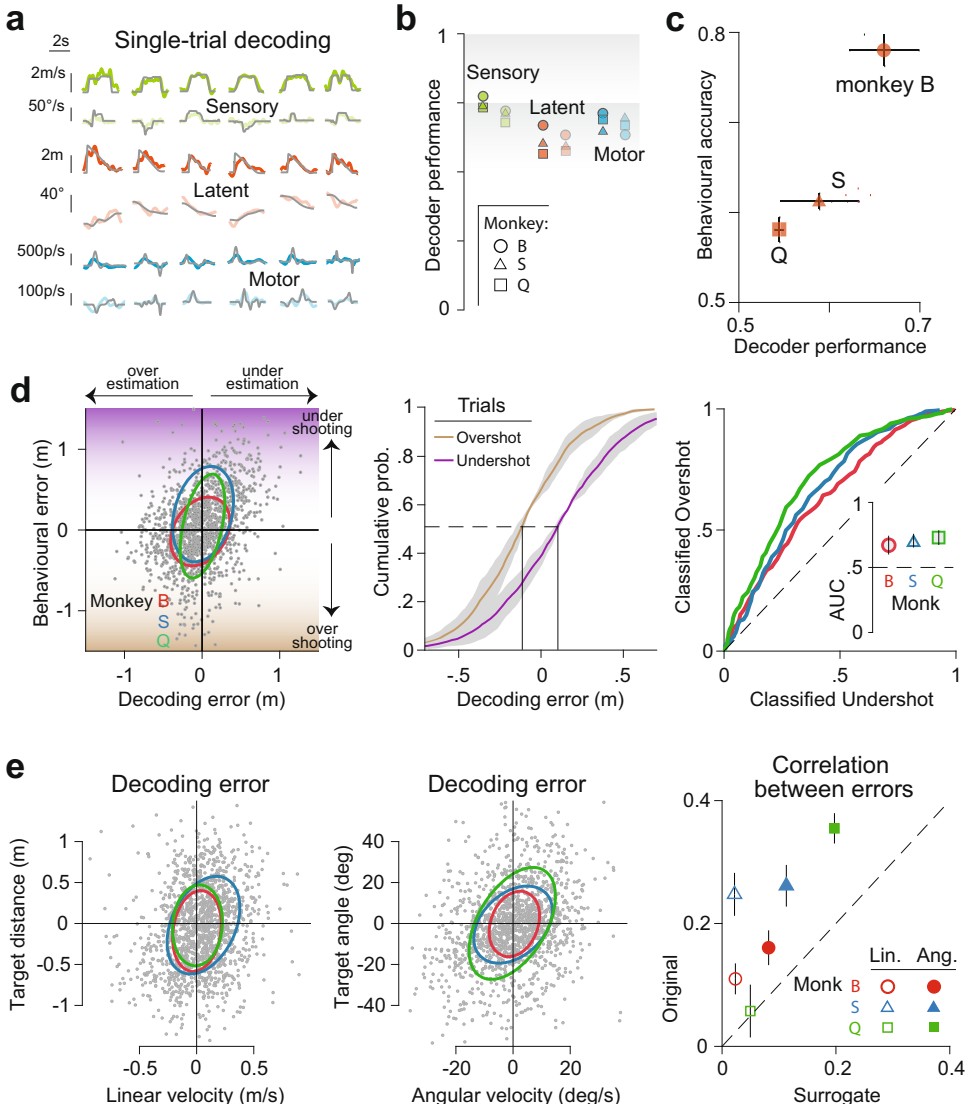

**Fig. 5 | Population activity predicts behavior on individual trials. a** We trained a linear decoder using ordinary least-squares to decode task variables from population activity (Methods). Columns correspond to six representative trials and rows correspond to different variables (linear velocity, angular velocity, target distance, target angle, and hand velocity along the two leading principal components of hand position). Gray traces correspond to ground truth. **b** Decoder performance for each variable, quantified as the correlation between observed and decoded estimates shown separately for individual monkeys. **c** Average performance of the decoder for distance-to-target plotted against the mean behavioral accuracy (fraction of correct trials) of different monkeys. **d** Left: Error in decoding distance to target is correlated with the behavioral error (difference between target distance and stopping distance) across trials. Middle: The cumulative distribution of decoding error shown separately for trials in which the monkey undershoots (purple) or

overshoots (gold). Right: ROC curves constructed by plotting the cumulative probabilities for the two sets of trials against each other, and the associated classification accuracy quantified as area under the curve (AUC, inset). **e** Left: Across trials (gray dots), the time-averaged error in decoding linear velocity is correlated with the time-averaged error in decoding linear velocity. Ellipses map out contours corresponding to 1 SD (assuming Gaussianity) for individual monkeys. Middle: Similar to left panel, but for the angular domain. Right: Linear (open symbols) and angular (closed symbols) correlation coefficients for individual monkeys computed from data, plotted against correlations computed using surrogate data. For each pair of decoders, surrogate correlations were computed by taking pairs of random projections of the original data along directions that overlap by the same angle as the pair of decoders. Error bars and shaded regions in **c**–**e** denote ± 1 SEM estimated by bootstrapping (n = 1000 trials). Source data are provided as a Source data file.

Fig. 5e−right). This suggests that neural computations include a significant interaction between activity subspaces representing sensory and latent variables. Such an interaction is consistent with a role of PPC in computing latent world states from sensory inputs.

## Effect of task manipulations

To better understand how neural computations for this task are implemented in monkey PPC, we tested how manipulating task variables affects neural representations. In separate sessions, two of the monkeys (S and Q) performed three variations of the baseline task in which we manipulated either the reliability of optic flow (sensory input) by changing the density of ground plane elements, the

consequence of actions (motor output) by changing the gain of the joystick, or the animal's position in the virtual world (latent state) by briefly dislodging them off their intended trajectory (details in Methods; Fig. 6a). Crucially, these sessions also included trials from the common, unmanipulated task such that we could record and contrast the activity of the same population of neurons under both conditions.

Behavior was robust to all three manipulations, but there was a small drop in performance (mean AUC ± SD across six sessions of each manipulation−baseline: 0.88 ± 0.03, average across manipulations: 0.84 ± 0.03, p = 0.002, paired t test; Figs. 6b and S6a). Because activity of each neuron was greatly influenced by the activity of other neurons, we first tested whether the shape of the coupling filters was affected by

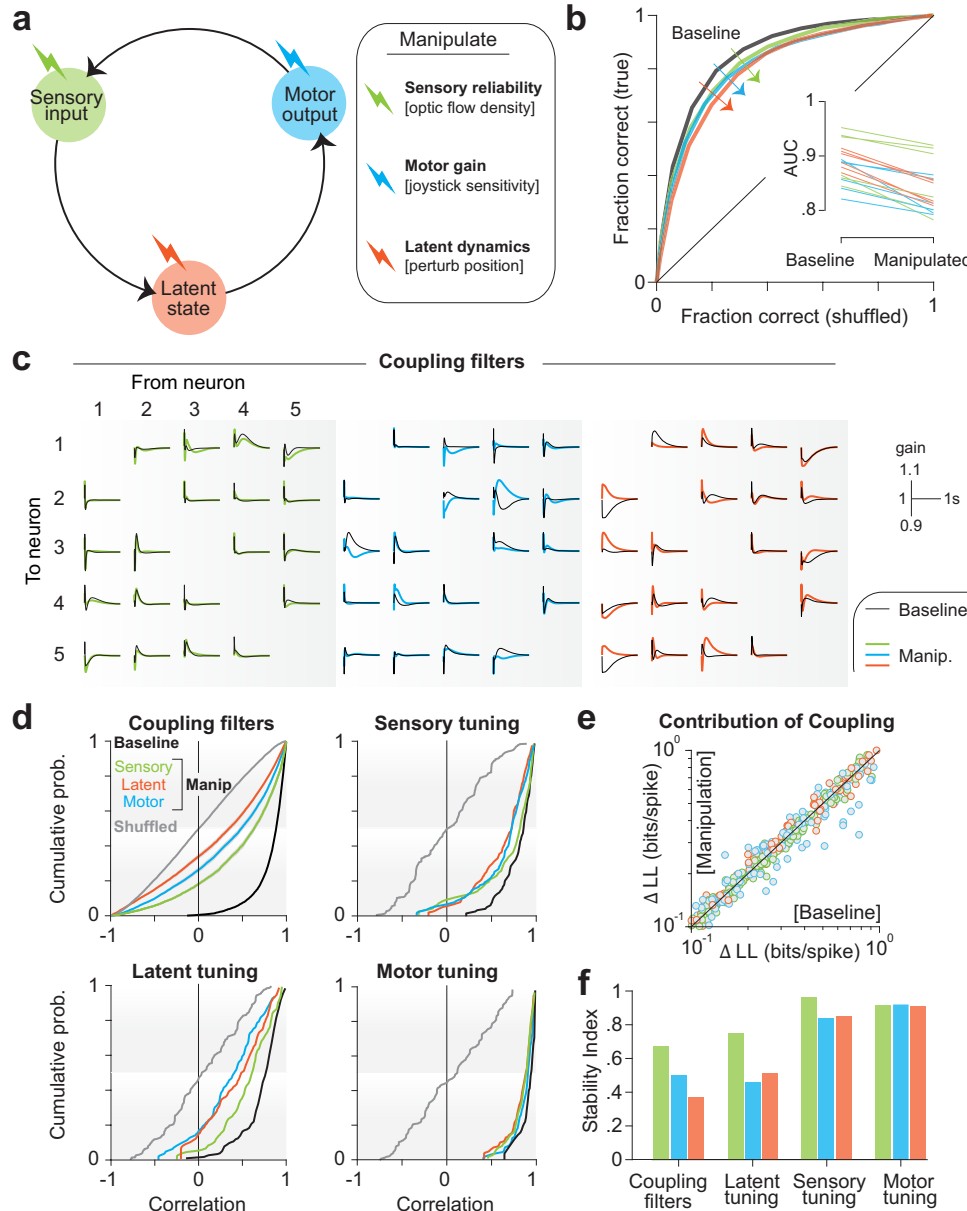

**Fig. 6 | Behavioral manipulations trigger changes in coupling and latent representation. a** In separate experimental sessions, we manipulated sensory input (by altering the density of optic flow), motor output (by altering the sensitivity of the joystick), or latent state (by perturbing the monkey off his intended trajectory while traveling) ("Methods"). **b** These manipulations produced modest behavioral effects as shown by the ROC curves for a subset of experimental sessions, quantified as area under the curve (AUC, inset). **c** Coupling filters (shown for a representative subset of all directed pairs of five neurons) fit to data from baseline trials (gray) and manipulation trials (colored). **d** Top left: Cumulative distribution across all pairs of neurons of the correlation coefficient between coupling filters fit to baseline trials and manipulated trials. Black curve shows the baseline distribution computed using odd and even baseline trials. Gray curves show the null distribution constructed by shuffling the neuronal pairs. Shaded regions denote ±1 SEM estimated by bootstrapping (n = 25,805/20,798/22,506 neuron pairs for sensory/

latent/motor manipulation). Top right: Cumulative distribution of the correlation coefficient between tuning function to sensory variables (the two sensory variables - linear and angular velocity - were concatenated) fit to baseline trials and manipulated trials across all neurons. Bottom left: Similar to top right, but computed using tuning to latent variables. Bottom right: Similar to top right, but computed using motor tuning. **e** Contribution of coupling to neuronal response, quantified as the improvement in the model log likelihood (LL) over uncoupled model, was comparable during baseline and manipulated trials. **f** Baseline-corrected stability to manipulated task (stability index) computed using cumulative distributions in **d** ("Methods") for coupling filters, latent tuning, sensory tuning, and motor tuning under all three manipulations. Both sensory and motor tunings were robust to manipulations, whereas coupling and latent tunings were not. Source data are provided as a Source data file.

task manipulations. We observed that all three manipulations altered coupling, albeit to varying degrees (Figs. 6c and S6b). For each pair of neurons, we quantified the degree to which coupling was altered by computing the correlation between filters fit to data recorded with and without task manipulation (Fig. 6d−top left in color). Under all manipulations, the median correlation dropped significantly below the noise ceiling defined as the correlation between couplings fit to odd/

even trials of the baseline task (color vs black), but remained significantly above chance level defined as the correlation between couplings between random pairs of neurons (color vs gray). Notably, manipulations did not change the extent to which coupling explained the activity of single neurons (paired $t$ test−sensory manip: $p = 0.81$, latent manip: $p = 0.68$, motor manip: $p = 0.23$; Fig. 6e). We quantified the stability of coupling filters to task manipulations by measuring the

stability index ("Methods") that ranged from 0 (highly unstable) to 1 (perfectly stable). According to the causal structure of the task, all three manipulations push the latent belief state about the world away from what the animal has come to expect based on his experience during the baseline task. However, for a given action, the latent world states in the sensory manipulation condition remain closer to the unmanipulated distribution. Strikingly, of the three manipulations, sensory manipulation produced the least change in coupling (Stability Index (SI) of coupling−sensory manip: $0.68 \pm 0.01$, latent manip: $0.36 \pm 0.01$, motor manip: $0.49 \pm 0.01$; Fig. 6f). This suggests that an internal model of the latent state dynamics may be embedded in the network connectivity, such that manipulations that defy the learned model are more likely to alter the interactions between neurons, either directly by re-calibrating the recurrent connections or by recruiting additional neural pathways, in order to maintain good behavioral performance.

If coupling filters reflect neural interactions that support the computation used to track the latent state dynamics by PPC neurons, then changes in coupling should produce changes in how single neurons represent the latent state. We tested this by computing the robustness of tuning functions to the task manipulations, and found that tuning to latent state (position) was indeed greatly affected (SI of latent tuning−sensory manip: $0.72 \pm 0.05$, latent manip: $0.53 \pm 0.03$, motor manip: $0.47 \pm 0.03$; Fig. 6d, f). In contrast, neuronal tuning to sensory and motor variables was relatively more stable (Figs. 6d, f and S6c). Thus, both sensory and motor representations were robust to task manipulations whereas latent state representation was not.

Taken together, results from the manipulation experiments suggest that the functional role of PPC in this task might be primarily to integrate sensory input (velocity) inherited from upstream brain areas in order to track the latent world state (position), which could then be used by downstream circuits to generate appropriate motor commands. Concretely, integration could be implemented locally via recurrent interactions within PPC which embody the world model, resulting in a dynamically changing representation of the latent world state in response to task manipulations. In contrast, sensory and motor signals may flow through relatively static input and output connections of PPC yielding more stable representations of those variables. To test the plausibility of this hypothesis, we trained a recurrent neural network (RNN) model to perform the same task as monkeys and compared the representation learned by the network to monkey PPC. The objective was to produce a model that was functionally equivalent to monkeys without explicitly constraining the representations it learned, such that any similarity between the representation of the RNN and the monkey PPC could be attributed to the computational constraints satisfied by the RNN.

## RNN model
We trained a fully connected RNN model (Fig. 7a) to solve a task similar to the one solved by monkeys. The network comprised 100 recurrently connected, nonlinear neurons whose activity ranged from −1 to +1. The network received four inputs (2D self-motion velocity and 2D target position) and the network activity was linearly read out by two controller output neurons (2D hand velocity). The control outputs affected the latent world states as well as the sensory inputs which were fed back via the two input channels conveying self-motion velocity (World model block; Methods). At the start of each trial, the network received transient pulses whose amplitude encoded the target position. We trained the network to produce controller outputs such that the resulting trajectory ended atop the target center ("Methods"). The network learned to generate qualitatively good trajectories (Fig. 7b−left), and the training was halted when the performance matched that of the monkeys (Fig. 7b−right, Fig. S7a).

Similar to monkey PPC neurons, model neurons exhibited sequential activity, with a precise sequence that was modulated by the goal location (Fig. 7c). However, model neurons were active for much longer periods and consequently exhibited lower sequentiality than monkeys' neurons (mean Sql ± 95% CI−Model: $0.08 \pm 0.05$, Monkeys: $0.29 \pm 0.1$). Introducing a metabolic constraint into the training objective by penalizing the average activity substantially improved the degree of sequentiality ("Methods"; Fig. S7b), suggesting that such constraints might be operating in PPC. Although the model was not explicitly trained to track the latent state (position), model neurons nonetheless exhibited tuning to target distance and angle (Fig. 7d−left) because this was needed to perform the task. There was no evidence for functional specialization and the model neurons exhibited a high degree of mixed selectivity to sensory, latent, and motor variables (Fig. S7c). Furthermore, we found that recurrent connectivity explained a large fraction of the activity (Mean $R^2 \pm 95\%$ CI−without coupling: $0.39 \pm 0.04$, with coupling: $0.92 \pm 0.1$). The distribution of couplings between neurons in the network reflected a balance between inhibitory and excitatory interactions (Fig. 7d−right, Fig. S7d). Similar to monkey PPC, the error in decoding the target distance from model neurons predicted the error in stopping position (Pearson's $r$: Model: $0.55 \pm 0.1$, Monkeys: $0.29 \pm 0.2$; Fig. 7e−left). This suggests that the latent state representation learned by the network propagates to the readout neurons that drive motor output. Likewise, the error in decoding the sensory input was correlated with the error in decoding the latent world states (Pearson's $r$−Linear velocity vs Target distance: $0.23 \pm 0.2$, Angular velocity vs Target angle: $0.45 \pm 0.2$; Fig. 7e−right) suggesting that the latent representations were derived, at least in part, by integrating sensory inputs. Finally, we tested the network on the three manipulations described in the previous section. Although the network was robust to adding more sensory noise, it generalized less well to latent state and motor manipulations (Fig. S7e). Network performance in the latter two manipulations could be restored by retraining the recurrent weights (Fig. 7f−top left, "Methods"). Notably, comparable performance in all three manipulations could be achieved without retraining the input weights or readout weights (Fig. 7f−top right). Consequently, tuning to both sensory and motor variables was stable under all manipulations (Fig. 7f−bottom). In contrast, tuning to latent state and the coupling between model neurons were only robust to sensory manipulation where the recurrent weights did not have to be retrained. These findings parallel the effect of manipulations on PPC neurons (Fig. 6f).

The striking similarity between the model and monkey neural representations suggests that the neural mechanisms by which PPC contributes to this task may be dictated largely by simple architectural constraints that were incorporated into the RNN model. Specifically, like the model neurons, PPC neurons might inherit self-motion information from upstream brain regions via stable pathways established during development. This could explain why tuning to self-motion velocity remains largely invariant under task manipulations. Analogous to the controller units in the model, downstream motor areas responsible for driving the muscles might decode the activity in PPC using fixed readout weights, yielding a stable relationship between PPC activity and hand movements. Finally, recurrent interactions within PPC might reflect knowledge of the world model, such that the stream of sensory inputs are filtered through those interactions to dynamically infer the latent world state. Critically, changes to the world model, such as those introduced here through task manipulations, would effectively modify the neural interactions and change the relationship between neural activity and the latent world state, as we see in the data.

## Discussion
To test whether neural circuits can track continuous and dynamical latent state variables, we designed a naturalistic paradigm in which monkeys navigated to a remembered goal location in a VR environment lacking explicit position cues. We recorded the activity of neurons in area 7a of the PPC and found that neurons exhibited sequential

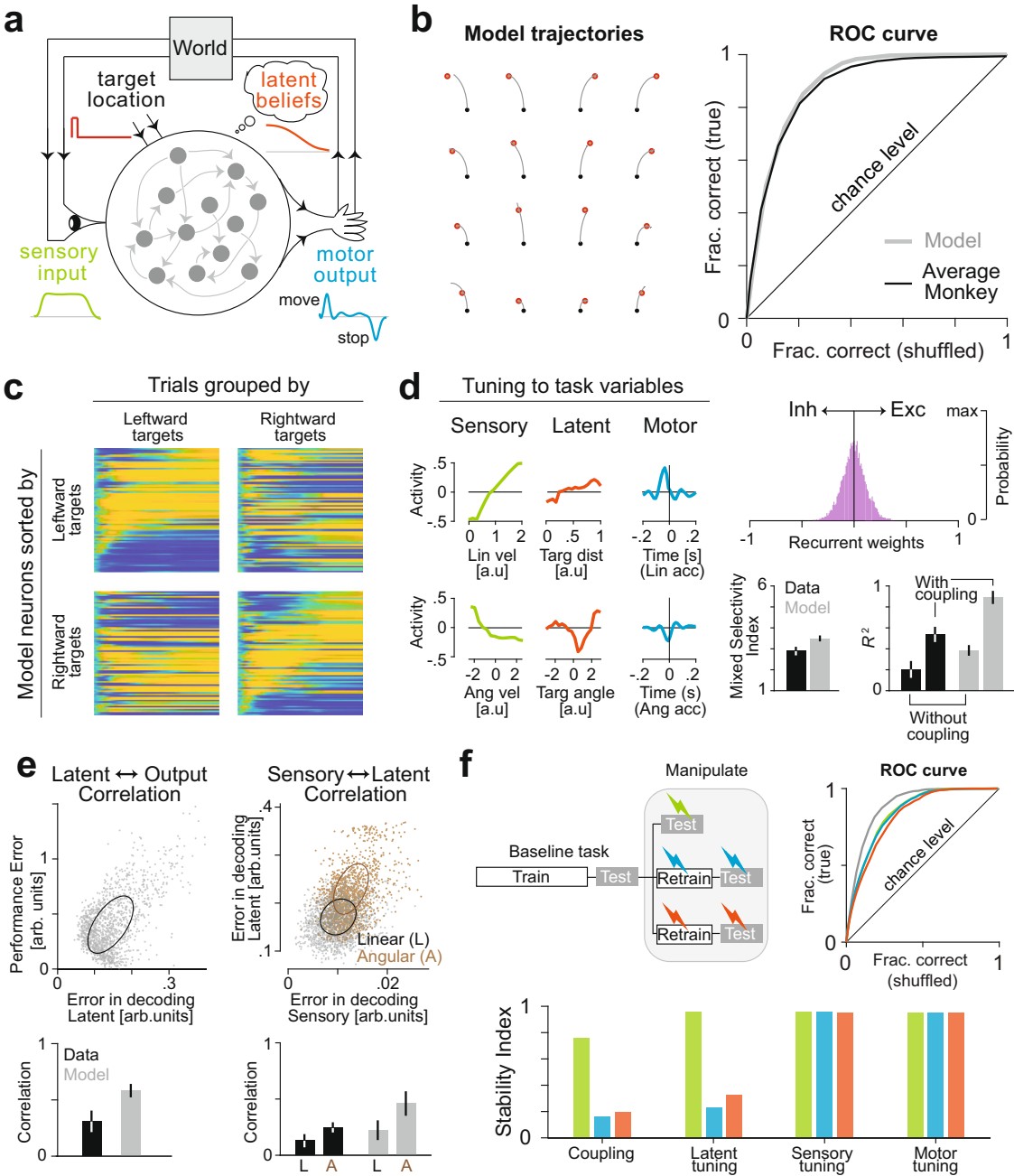

activity at the population level, were strongly influenced by other neurons, and encoded latent states at the neuronal level. Latent states decoded from population activity on individual trials were correlated with the monkeys' stopping position relative to the goal. Finally, manipulating sensory reliability, latent state, or motor gain altered functional coupling between neurons and also affected the latent state representation, but spared sensory and motor representations.

A large body of work in primates and rodents has found evidence that PPC neurons encode a dynamic decision variable when inferring latent causes from sensory inputs[8,11], and that the neural activity predicts behavior[9]. We highlight three ways in which the paradigm used here differs from past studies and why it matters. First, in standard paradigms such as motion discrimination or the towers task[8,37], the animal integrates evidence in favor of a categorical proposition. The latent state itself is discrete and time-invariant, and the integration process serves to average away the noise in sensory input thereby improving decision confidence. In contrast, the latent state in the paradigm used here (relative goal location) is both continuous and

time-varying, such that integration is needed to continually track this dynamical state. This does not obviate the need to gather momentary evidence about self-motion from noisy sense data. Instead, monkeys had to perform both computations - infer movement velocity from optic flow, and then integrate that to track the latent world state[34]. Second, in contrast to binary decision-making paradigms, the task used here allows for a continuous behavioral readout via joystick movements and a greater decoupling of latent state variables and motor output. Third, due to the interactive nature of this task, spatiotemporal statistics of the sensory input is not predetermined by the experimenter, but generated online by the monkeys' own actions, mimicking the closed-loop nature of real-world behaviors. This paves the way for a more direct comparison with neural activity in rodents, where such interactive behaviors are more commonly studied. At the same time, this did not entail sacrificing experimental control as we could independently manipulate sensory reliability (optic flow density), latent state (position), and the motor plant (joystick gain) and examine their consequences on behavior and neural response. For

**Fig. 7 | A recurrent neural network model operating in closed loop recapitulates experimental findings. a** Schematic of the recurrent neural network architecture. The network comprised 100 fully-connected neurons receiving a transient signal (target location) on two input channels, and two motor output channels that control linear and angular acceleration which then determined the signal received by two sensory input channels (velocity). Analogous to the function of the virtual-reality setup in our experiments that converts the joystick output into visual input, the "World" block integrates the motor output to generate subsequent velocity input. So this architecture mimics the interaction between the monkey and the virtual reality. **b** The recurrent weights were trained using a standard supervised learning algorithm (Backpropagation-through-time (BPTT), "Methods") to generate appropriate outputs to random target locations. Left: The network's trajectory in response to 16 different targets are shown (red−target, black−starting location). Right: Learning was stopped when the network performance matched the average monkey as measured by the respective ROC curves. **c** Neurons in the network exhibited sequential dynamics in a target location dependent manner (compare with Fig. 2). **d** We fit a generalized additive model (GAM) to the model neurons. Left: Model neurons were tuned to different task variables including latents (despite not explicitly training to learn them) (compare with Fig. 3). Right: Coupling was weak and concentrated around zero (top) but nonetheless affected the goodness of fit (bottom-right) (compare with Fig. 4). Model neurons exhibited mixed selectivity to

a degree that was comparable to PPC neurons (bottom-left, "Methods"). Error bars denote ±1 SEM ($n = 244/n = 100$ for data/model). **e** We trained linear readouts on the model population response to decode task variables. Left: Across trials (dots), the error in decoding target distance was correlated with the performance (quantified by stopping distance) of the network (compare with Fig. 5d). Right: Errors in decoding target distance and angles were correlated with errors in decoding linear (gray) and angular velocity (brown), respectively (compare with Fig. 5e). **f** Simulated manipulations. We added different amounts of noise to the sensory input channels (sensory manipulation), multiplied the transformation implemented by the "World" block by a non-unity gain factor (motor manipulation), or added a randomly timed Gaussian pulse to the sensory input channels to displace the model off its intended trajectory (latent manipulation). Top: The network readily adapted to sensory manipulation without additional training, but motor and latent manipulations required a small amount of additional training of recurrent weights to elicit comparable behavioral performance. Bottom: Sensory and motor tunings were robust to all three manipulations largely because the input and output weights did not change during the additional training. However, similar to PPC neurons (compare with Fig. 6f), the coupling and latent tunings were affected because the additional training modified the recurrent weights and thus also the latent state representation. Source data are provided as a Source data file.

these reasons, the paradigm used here helps generalize past findings toward the domain of natural behavior, and sheds new light on the computational role of PPC.

At a very high level, the task required storing and manipulating latent states in working memory. Classic working memory tasks have reported two qualitatively different types of neural activity dynamics depending on the task: persistent activity[38,39] and sequential activity[9,40]. Few neurons exhibited activity that persisted throughout the trial in our task, but there was robust sequential activity at the population level. A recent study in rodent PPC proposed that sequential activity observed during virtual navigation might simply be inherited from the input due to the presence of spatial cues that signaled location[41]. Because there were no explicit landmarks in our task, our results demonstrate that structured inputs are not needed for sequential dynamics in parietal cortex. Our findings agree with a recent computational study on task-optimized neural networks which suggests that working memory tasks with greater complexity favor sequential dynamics, whereas simpler tasks such as delayed recall favor persistent activity dynamics[42]. That same study also demonstrated that the strength of coupling between neurons was greater in networks with sequential activity. Indeed, we found that coupling between neurons contributed substantially to the activity of individual neurons (>80% increase in information about precise spike times predicted with versus without coupling). In contrast, previous work in monkeys showed that coupling had a more modest effect (~20% increase) during a memory-guided saccade task that elicited persistent activity in lateral intraparietal area (LIP) neurons[43]. This difference persisted even when we subsampled our population and thus could not be attributed to the larger population size of our recordings. Our results are quantitatively more compatible with the strong coupling reported in PPC of mice performing an evidence accumulation task in VR where trial lengths were much longer[44], suggesting that task complexity might be the primary determinant of neural activity dynamics in PPC. One possibility is that persistent activity can support holding information in memory for a brief period (i.e., short-term memory[45]), but manipulating the information content in working memory (e.g., updating the latent state) might require richer dynamics such as sequential activity.

At the neuronal level, we found that about one-third of the neurons were tuned to the latent state, which here was the position of the monkey relative to the goal. This is similar to a recent finding that PPC neurons in cats trained to step over obstacles on a treadmill, encoded distance to those obstacles[46]. We did not observe ramping activity dynamics such as those observed in monkey area 7a and LIP during motion discrimination tasks, which can alternatively be interpreted as

tuning to net evidence. As noted by others, the tuning need not be monotonic to support evidence accumulation[47,48] and so ramping dynamics might emerge only under restricted settings. Precisely which settings favor ramping over other solutions should become more clear once the link between neural dynamics and computation are fully elucidated[49]. Additionally, roughly half of the PPC neurons were tuned to sensory (linear and angular velocity) and motor (hand movement) variables. Mixed selectivity to task variables has been widely documented in many cortical areas[50,51] including rodent PPC[52]. Non-linear mixed selectivity has been argued to increase neural dimensionality, thereby allowing linear readout mechanisms to solve arbitrary classification problems[53]. This computational benefit might also extend to regression problems, such as ours, where a representation with high expressivity would allow PPC to generalize better to new task variations[54].

Decoding analysis revealed that all task-relevant variables could be dynamically decoded from the population activity to a high degree of precision. Of note is the finding that trials in which the decoder underestimated (or overestimated) the target distance tended to result in undershooting (or overshooting) by the monkey. Recall that monkeys stop where they believe the target is located. Therefore, this result directly links PPC neural activity to the monkeys' belief about a continuous-valued, dynamical latent state. Relationships linking neural activity and continuous variables have been found for several sensory (e.g., orientation, motion direction) and motor (e.g., hand position, running speed) variables, but for latent states, such relationships are almost always quantified by measuring the correlation between neural activity and a binary choice[55–57]. This measure is an artifact of using experimental paradigms that do not allow for a direct readout of the animal's estimate of the latent state, but only an indirect readout after that estimate undergoes further nonlinear processing that affects decision-making, such as thresholding. In contrast, the analysis used here shows that trial-by-trial fluctuations in the neural state of PPC is in fact correlated with the monkeys' continuous internal estimate, and reveals the value of this experimental paradigm in establishing a tighter link between neural activity and latent beliefs. Since latent beliefs are formed by integrating past sensory inputs, a role in latent state computation also provides a parsimonious account of recent findings that PPC guides behavior based on sensory-history[58,59].

The above finding suggests that the latent state representation in PPC propagates to behavior, but is the latent state information inherited from elsewhere in the brain? We found signatures of latent state computation within PPC. Specifically, the error in decoding sensory variables (linear and angular velocity) was correlated with the error in

decoding latent state variables (target distance and target angle). Such a correlation would not arise if the information about latent states and sensory inputs were inherited from independent sources, but is entirely expected if latent states were computed by integrating noisy sensory input within a network that includes PPC. It is theoretically possible that both sensory and latent state variables are acquired from a common brain area. Testing this would require causal intervention experiments and brain-wide recordings, which we hope to perform in the future. Meanwhile, we assumed that PPC contributed to the computation of the latent states, and probed the underlying neural mechanisms using experimental manipulations.

All three experimental manipulations yielded strikingly similar effects. They altered both neuronal coupling and the latent state representation, but left sensory and motor tuning unaltered. While previous studies have characterized how functional connectivity between brain-wide networks changes across tasks[60,61] and across epochs within a task[43], how task manipulations affect coupling between neurons locally within a brain area is not known. Our results suggest that coupling can vary with task manipulations and identifies a potential mechanism by which behavioral performance in this task can be maintained in the face of external perturbations to task variables—namely, by reconfiguring the neural interactions within PPC. This reconfiguration could either result from synaptic modifications of recurrent connections within PPC or from contextual inputs to PPC that alter the effective interactions[62]. Either way, the concomitant changes observed in latent tuning suggest that latent states are likely computed via recurrent mechanisms within PPC, an interpretation that was further validated by the striking resemblance with the representations learned by a recurrent neural network (RNN) model trained to perform the same task. Finally, robustness of sensory and motor tuning to task manipulations was also readily explained by the same model by freezing the input and output weights and retraining only the recurrent weights. This suggests that sensory/motor variables inhabit a stable subspace in PPC, and could be transmitted from/to other brain areas via stable communication channels. It follows that the communication subspace between PPC and sensory cortex, as well as between PPC and motor cortex should be remain relatively invariant to context, a prediction that future studies can test using available tools[63]. PPC has been variously described as a brain area that is involved in sensory processing, working memory, and motor planning. Our results suggest broadening existing views: PPC computes and continuously monitors the dynamical latent state of the organism in naturalistic behaviors involving action/perception loops. This is in line with the perspective of PPC as a state estimator for optimal feedback control to flexibly interface sensory information with actions[64,65].

A limitation of our treatment of the neural computations in this task is that we have overlooked the contribution of autonomous strategies. In principle, it is possible to use the learned world model to plan joystick movements ahead of time without any sensory feedback. Although we know from manipulation experiments that monkeys relied on sensory feedback (optic flow), the contribution of autonomous strategies to navigation is likely non-negligible[35,66,67]. Planning-based computations, which require reasoning about the consequences of action sequences via mental simulation, are thought to be performed in the prefrontal cortex (PFC)[68,69]. Predictive signals that enable planning in dynamic environments have also been found recently in the dorsal anterior cingulate cortex[70]. A promising direction for future could be to compare the causal contributions of PPC against frontal brain regions in this task, by combining inactivation experiments and statistical tools that characterize behavior in the spectrum between purely sensory feedback-based vs purely autonomous strategies. Another open question concerns the stability of representations across time. Previous studies have demonstrated that neural representations in the rodent PPC drift over time[71]. While we have analyzed how representations are affected by task manipulations,

knowing whether they are stable across long timescales would complement the insights gained from this study and help further constrain the underlying mechanisms.

## Methods

### Experimental model
Three rhesus macaques (*Macaca mulatta*) (all male, 7-8 years. old)—referred to as B, S, and Q for simplicity—participated in the experiments. All surgeries and experimental procedures were approved by the Institutional Review Board at Baylor College of Medicine, and were in accordance with National Institutes of Health guidelines.

### Experimental setup
Monkeys were chronically implanted with a lightweight polyacetal ring for head restraint, and scleral coils for monitoring eye movements (CNC Engineering, Seattle WA, USA). Utah arrays were chronically implanted in area 7a of the left hemisphere of all three monkeys using craniotomy. Prior to the surgery, the brain area was identified using structural MRI to guide the location of craniotomy. After craniotomy, the array was pneumatically inserted after confirming the co-ordinate of the target area using known anatomical landmarks. The electrode arrays implanted in monkeys Q and B were composed of a $10 \times 10$ grid of 96 silicon microelectrodes, each 1 mm long and spaced 400 μm apart. The array implanted in monkey S had identical electrode lengths and spacing, except that it was composed of a $6 \times 8$ grid of 48 microelectrodes. At the beginning of each experimental session, monkeys were head-fixed and secured in a primate chair placed on top of a platform (Kollmorgen, Radford, VA, USA). A 3-chip DLP projector (Christie Digital Mirage 2000, Cypress, CA, USA) was mounted on top of the platform and rear-projected images onto a $60 \times 60$ cm tangent screen that was attached to the front of the field coil frame, ~30 cm in front of the monkey. The projector was capable of rendering stereoscopic images generated by an OpenGL accelerator board (Nvidia Quadro FX 3000G).

### Virtual reality
Monkeys used an analog joystick (M20U9T-N82, CTI electronics) with two degrees of freedom and a circular displacement boundary to control their linear and angular speeds in a virtual environment. Fore-aft and sideways movement of the joystick controlled linear and angular velocity respectively. The virtual world comprised a circular ground plane with a radius of 70 m (near and far clipping planes at 0.05 m and 40 m respectively), with the subject positioned at its center at the beginning of each trial. The ground plane was textured with small isosceles triangles (base × height: 0.85 cm × 1.85 cm) that were each randomly repositioned and reoriented anywhere in the arena at the end of its limited lifetime (~250 ms), making them impossible to use as landmarks. The maximum linear and angular speeds were fixed to 2 m/s and 90 °/s respectively, and the density of the ground plane was fixed at 2.5 elements/m². The stimulus was rendered as a red-green anaglyph and projected onto the screen in front of the subject's eyes. Monkeys wore goggles fitted with Kodak Wratten filters (red #29 and green #61) to view the stimulus. The binocular crosstalk for the green and red channels was 1.7% and 2.3%, respectively.

### Behavioral task
In each session, monkeys performed a series of trials in which they had to steer to a random target location that was cued briefly at the beginning of the trial. Each trial was programmed to start after a variable random delay (truncated exponential distribution, range: 0.2–2.0 s; mean: 0.5 s) following the end of the previous trial. The target, a circular disc of radius 20 cm whose luminance was matched to the texture elements appeared at a random location between $\theta = \pm 40°$ of visual angle at a distance of $r = 0.7$–4 m relative to where the subject was stationed at the beginning of the trial. The target only appeared

transiently on the screen for 300 ms, but the joystick was always active so monkeys were free to start moving before the target vanished. Monkeys were teleported to the origin of the environment at the time of target onset. Since the environment was comprised of flickering triangles, teleportation was seamless and did not interfere with the continuous nature of the task. Trials were aborted after a maximum duration of 7 seconds (5 seconds in a subset of sessions). Monkeys typically performed a block of ~1500 trials in each experimental session, and received binary feedback following a variable waiting period after stopping (truncated exponential distribution, range: 0.1–0.6 s; mean: 0.25 s). They received a drop of juice if their stopping position was within 0.6 m away from the center of the target. No juice was provided otherwise. Monkeys were first trained extensively, gradually reducing the size of the reward zone until their performance stopped improving. In this study, we focus only on their post-training behavior. At this point, the radius of the reward zone was fixed across trials. The fixed reward boundary of 0.6 m was determined during pilot experiments using a staircase procedure to ensure that monkeys received reward in approximately two-thirds of the trials. We collected behavioral data from 110 recording sessions (27 from monkey B, 18 from monkey S, and 65 from monkey Q) yielding a total of 121,930 trials for behavioral analyses.

## Task manipulations

We performed three different task manipulations on monkeys S and Q. One of these involved manipulating the reliability of the sensory observations (optic flow) by changing the density of the ground plane element. Trials with two densities that differed by a factor of 25 (2.5 elements/m² and 0.1 elements/m²) were randomly interleaved in these sessions. In a second version, we manipulated the effect actions (hand movements) inflicted on the latent state by altering gain of the joystick controller. In these sessions, we interleaved trials in which the gain of the joystick controller was switched between 1× (identical to baseline), 1.5×, and 2×. Within each trial, both linear and angular velocities were scaled by the same gain factor in order to avoid inducing different effects on linear and angular responses. Finally, we disrupted the transitions between the latent states by adding a brief external passive displacement that moved the subjects away from their expected path, at a random time during the trial. The perturbations had a fixed duration of 1 s and their velocity had a Gaussian profile with a standard deviation of 0.2 s and an amplitude that, on each trial, was drawn randomly from a uniform distribution bound between −2 and 2 m/s and between −120 and 120 °/s for the linear and angular velocity, respectively. The perturbation onset time was randomly varied from 0 to 1 s after movement onset.

## Behavioral recording and acquisition

All stimuli were generated and rendered using C++ Open Graphics Library (OpenGL) by continuously repositioning the camera based on joystick inputs to update the visual scene at 60 Hz. The camera was positioned at a height of 0.1 m above the ground plane. Spike2 software (Cambridge Electronic Design Ltd., Cambridge, UK) was used to record and store the time series of target locations as well as the animal's location in the virtual environment for offline analysis. All behavioral data were recorded along with the event markers at a sampling rate of $833\frac{1}{3}$ Hz.

## Tracking of eye and hand movements

We recorded the horizontal and vertical positions of both eyes using chronically implanted scleral search coils in monkeys Q and **B**. Eye-tracking in monkey S was performed using a video-based eye-tracking system (ISCAN Inc., Woburn, MA, USA). Additionally, a video of the monkeys' hand movements was captured at 30 frames/s using a 1280 × 960 (1.2 Megapixels) industrial-grade monochrome CCD camera (DMK 23U445, The Imaging Source LLC, Charlotte, NC, USA). The start and

end of the video recording was synchronized with other behavioral data using a trigger-pulse sent by the stimulus acquisition software (Spike2). We used *DeepLabCut*[72], a Python toolbox, to extract the trajectory of hand movements from the above videos. To do this, we first labeled the same set of identifiable features (fingers and wrist) in a random subset of 200 frames from one randomly chosen video recording. We then trained a deep neural network model using *DeepLabCut* on an NVIDIA Quadro P5000 GPU until the training error for the set of labeled frames saturated (typically around 500,000 iterations). Finally, we analyzed all the videos using the trained network to extract the time course of the spatial location of the features of interest.

## Neural recording and acquisition

We recorded extracellularly using multi-electrode arrays (Blackrock Microsystems, Salt Lake City, UT, USA) from area 7a. Broadband neural signals were amplified and digitized at 30 KHz using a digital headstage (Cereplex E, Blackrock Microsystems, Salt Lake City, UT, USA), processed using the data acquisition system (Cereplex Direct, Blackrock Microsystems) and stored for offline analysis. Additionally, for each channel, we also stored low-pass filtered (−6 dB at 250 Hz) local-field potential (LFP) signals sampled at 500 Hz. Finally, copies of event markers were received online from the stimulus acquisition software (Spike2) and saved alongside the neural data.

## Spike detection and sorting

Spike detection and sorting were initially performed on the raw (broadband) neural signals using MATLAB *KiloSort*[73] software on an NVIDIA Quadro P5000 GPU. The software uses template-matching both for detection and clustering of spike waveforms. The spike clusters produced by *KiloSort* were visualized with a Python package called *Phy* and manually refined by a human observer using standard heuristics. A typical recording session yielded 70–100 neurons across electrodes.

## Models

**Generalized additive model.** To test whether task variables modulate neural activity, we fit a Poisson generalized additive model (GAM) to the responses of individual neurons. The model relates spike counts of $\mathbf{r}_t \in \mathbb{Z}_+^N$ of the neural population to continuous-valued input variables $\mathbf{x}_t \in \mathbb{R}^{N_C}$, binary events $\mathbf{z}_t \in \{0,1\}^{N_E}$ and past neural activity $\mathbf{r}_{1:t-1}$ according to:

$$\log(\mu_t^i) = \sum_{k=1}^{N_C} f_k^i(x_t^k) + \sum_{l=1}^{N_E} (g_l^{i*}z_{1:t-1}^l) + (h^{i*}r_{1:t-1}^i) + b^i \qquad (1)$$

where $r_t^i \sim \text{Poisson}(\mu_t^i)$ denotes the Poisson-distributed response of neuron $i$ at time $t$, $x_t^k$ is magnitude of the $k^{th}$ continuous-valued input variable at time $t$, $f_k^i(\cdot)$ is any generic nonlinear function operating on $x^k$, $z_t^l$ is the value of the $l^{th}$ binary event at time $t$, $g_l^i$ is the temporal filter operating on $z^l$, $h^i$ is the causal spike-history filter that accounts for the refractory period and other autoregressive effects, $N_C$ & $N_E$ denote the total number of continuous-valued inputs and binary events respectively, '*' denotes the convolution operator, and $b^i$ is an additive constant to capture tonic firing. This model did not take recurrent interactions between into account, so we refer to it as the **uncoupled** model. We also fit an extension of the above model that included coupling between neurons as follows:

$$\log(\mu_t^i) = \sum_{k=1}^{N_C} f_k^i(x_t^k) + \sum_{l=1}^{N_E} (g_l^{i*}z_{1:t-1}^l) + (h^{i*}r_{1:t-1}^i)$$
$$+ \sum_{j=1,j\neq i}^{N} (p_j^{i*}r_{1:t-1}^j) + b^i \qquad (2)$$

where $p_j^i$ is the causal coupling filter that captures the directional interaction from neuron $j$ to neuron $i$ and $N$ denotes the total number of neurons in the recording. We refer to this as the **coupled** model. Details about model parameters are stated in the Model fitting section.

**Recurrent neural network model.** We trained a fully connected recurrent neural network (RNN) comprising $N = 100$ nonlinear firing rate units to solve the same task as the monkeys. The network contained $M = 4$ input channels, two for conveying the 2D target location (**x**) encoded in the amplitude of a transient pulse delivered in the beginning of the trial and two for conveying sensory feedback about the 2D self-motion velocity (**z**) throughout the trial. There were $P = 2$ output channels, one each for controlling the velocity of the 'hand' along the linear and angular axes of the joystick (**y**). The network was similar to those commonly trained to solve standard neuroscience tasks, but with one key architectural modification: the output channels were temporally integrated and fed back to the network through the input channels conveying movement velocity (i.e., $\mathbf{z}_t = \int_0^t \mathbf{y}_s \, ds$), thereby closing the sensorimotor loop. This feedback mimics the functionality of the virtual reality simulator that uses the joystick output to render real-time sensory feedback in the form of optic flow in our experiments. To mimic noise in the motor periphery, we added a small amount of process noise to the output channels before integrating. The equation governing the network dynamics was:

$$\tau \dot{\mathbf{r}} = -\mathbf{r} + (W^{\mathrm{rec}}\mathbf{r} + W^{\mathrm{in}}\tilde{\mathbf{x}}) \quad \text{and} \quad \mathbf{y} = W^{\mathrm{out}}\mathbf{r} \tag{3}$$

where **r** is the population activity, $\dot{\mathbf{r}}$ denotes its time-derivative, **y** is the network output representing hand velocity, $\tilde{\mathbf{x}} = (\mathbf{x}, \mathbf{z})$ denotes the input to the network obtained by concatenating the target location and sensory feedback obtained by integrating the network output, $\tau$ is the cell-intrinsic time constant, and $(\cdot) = \tanh(\cdot)$ is the neuronal nonlinearity. Matrices $W^{\mathrm{rec}} \in \mathbb{R}^{N \times N}$, $W^{\mathrm{in}} \in \mathbb{R}^{N \times M}$ and $W^{\mathrm{out}} \in \mathbb{R}^{P \times N}$ correspond to recurrent, input, and output weights respectively. Details about inputs, outputs, and the training procedure used to learn the network parameters are stated in the Model fitting section.

**Linear decoder.** For each recording session, we regressed the time course of population pattern of instantaneous firing rates $R \in \mathbb{R}^{T \times N}$ (where $N$ is the size of the neural population and $T$ is the total number of time bins) separately against each continuous-valued variable $\mathbf{x}_{1:T}$ to obtain weights $\mathbf{w} = (R^{\mathrm{T}}R)^{-1}R^{\mathrm{T}}\mathbf{x}$. Firing rates were estimated by convolving the spike train with an exponential filter with time constant $\eta$ as a hyper-parameter. For each target variable, we obtained the regression weights **w** using data from the training set (80% trials) and decoded that variable from the population activity observed in a validation set (10% trials) to estimate the decoding error $\epsilon = \sqrt{\sum_t (\mathbf{w}^{\mathrm{T}}R_t - x_t)^2}$ where $t$ denotes time bin. For each task variable, we determined the optimal timescale of the filter $\eta$ within a range between ~25 and 250 ms as the timescale that minimized the decoding error in the validation set. Finally, decoding performance was evaluated by decoding the population activity observed an independent test set (remaining 10% trials) using regression weights corresponding to the optimal timescale.

## Model fitting and evaluation

**Generalized additive model.** Given a time series of neuronal responses **r** of a population of neurons, and inputs **x** & **z**, the goal is to recover the set of all tuning functions $\mathbf{f}^i$, temporal filters $\mathbf{g}^i$, $\mathbf{h}^i$, coupling filters $\mathbf{p}^i$ and the additive constant $b^i$ for each neuron $i$. We solve this by

computing the maximum a posteriori (MAP) estimate:

$$\{\hat{\mathbf{f}}^i, \hat{\mathbf{g}}^i, \hat{\mathbf{h}}^i, \hat{\mathbf{p}}^i, \hat{b}^i\} = \mathbf{f}^i, \mathbf{g}^i, \mathbf{h}^i, \mathbf{p}^i, b^i \arg\max P(\mathbf{r}, \mathbf{x}, \mathbf{z} | \mathbf{f}^i, \mathbf{g}^i, \mathbf{h}^i, \mathbf{p}^i, b^i) P(\mathbf{f}^i, \mathbf{g}^i, \mathbf{h}^i, \mathbf{p}^i, b^i) \tag{4}$$

where $P(\mathbf{r}, \mathbf{x}, \mathbf{z} | \mathbf{f}^i, \mathbf{g}^i, \mathbf{h}^i, \mathbf{p}^i, b^i) = \prod_t e^{-\mu_t^i}(\mu_t^i)^{r_t^i}/r_t^i!$ is the model likelihood for the $i$th neuron where $\mu_t^i$ is given by Eq. (1), and $P(\mathbf{f}^i, \mathbf{g}^i, \mathbf{h}^i, \mathbf{p}^i, b^i)$ is the prior over model parameters. We chose a factorizable Gaussian prior on the curvature of the tuning functions $\mathbf{f}^i$, the temporal filters $\mathbf{g}^i$ and the spike-history filter $\mathbf{h}^i$ to encourage smoothness, a Laplace prior on coupling filters $\mathbf{p}^i$ to encourage sparseness, with no prior constraints on $b^i$:

$$P(\mathbf{f}^i, \mathbf{g}^i, \mathbf{h}^i, \mathbf{p}^i, b^i) = \prod_{j=1}^{N} \prod_{l=1}^{N_E} \prod_{k=1}^{N_C} P(f_k^i)P(g_l^i)P(h^i)P(p_j^i) = \prod_{j=1}^{N} \prod_{l=1}^{N_E} \prod_{k=1}^{N_C}$$

$$\exp\left\{ -\lambda_k \left|\frac{\partial f_k^i}{\partial x^k}\right|^2 - \gamma_l \left|\frac{\partial g_l^i}{\partial t}\right|^2 - \alpha \left|\frac{\partial h^i}{\partial t}\right|^2 - \beta |p_j^i| \right\}$$

where $\lambda_k$, $\gamma_l$, $\alpha$, and $\beta$ are the hyperparameters that penalize rough tuning functions, rough temporal kernels, and dense coupling. $\|\cdot\|$ and $|\cdot|$ denote the $\ell_2$ norm and the $\ell_1$ norm respectively. After fitting the model parameters, we estimated the marginal tuning functions $\mathbb{E}[\hat{r}^i | x^k]$ to each variable $x^k$ by computing the conditional expectation of model-predicted response $\hat{r}^i$ given variable $x^k$ by marginalizing over the remaining variables:

$$\mathbb{E}[\hat{r}^i | x^k] = e^{f_k^i}\left(\prod_{\substack{j=1 \\ j \neq k}}^{N_C} \int e^{f_j^i(x^j)}P(x^j)dx^j\right)\left(\prod_{l=1}^{N_E} \int \frac{1}{T}e^{g_l^i(t)*z^l(t)}dt\right) \tag{5}$$

where we have assumed that the joint probability density function over task variables can be factorized into a product of marginal densities. Under this assumption, tuning to each task variable is multiplicatively modulated by the remaining task variables without affecting its shape. Furthermore, we ignored the effect of spike-history and coupling filters because these filters did not substantially affect the average firing rate of the neuron predicted by the model i.e., their multiplicative modulation was close to unity. Marginal temporal responses to events $z^l$ were determined by computing $\mathbb{E}[\hat{r}^i | z^l]$ in an analogous fashion. To fit the model using experimental data, we used different combinations of $N_C = 9$ continuous-valued variables—two sensory variables (linear velocity and angular velocity), two internal estimates (distance to target and target angle), two motor variables (hand speed along the first two principal components of hand position), the instantaneous phase of the local field potential (LFP), and the two components of eye position (horizontal and vertical)—and $N_E = 2$ discrete events (target onset and reward onset). Although the motor variables (hand speeds) were continuous-valued, they changed in a phasic manner with most changes concentrated around the onset of navigation and end of navigation. Preliminary analyses indicated that the associated neural changes were better captured by (acausal) temporal filtering of hand speed than tuning functions to hand speed. Therefore, we fit temporal kernels to capture the relationship between the motor variables and neuronal activity.

To fit the functions **f**, **g**, **h**, **p**, we expressed each of them as a linear combination of basis functions. Tuning functions **f** were parameterized using a basis of ten boxcar functions, where each function spanned an equal range of the predictor variable. Temporal filters **g** were parameterized using a basis of ten raised cosine filters spanning a range of 600 milliseconds. The filter associated with target-onset was causal ([0, 600] ms), while the remaining filters were non-causal ([-300, 300] ms). Both spike-history filter **h** and coupling filter **p** were expressed using a basis of ten causal raised cosine filters in logarithmic

time scale. Spike-history filters spanned 350 ms, while coupling filters spanned 1.375 seconds. For each category of filter, the time duration was set to be the largest value beyond which the filters did not substantially improve the model likelihoods in preliminary analyses.

Regularization hyperparameters were first determined using a cross validation procedure on a subset of neurons. In this procedure, we varied the hyperparameter values on a logarithmic scale from 0.001 to 1000 and fit the model by including all task variables for each hyperparameter setting using 90% of the data, and chose the hyperparameter combination with the highest model likelihood in the remaining 10% of the data. To reduce the complexity of this procedure, we assumed a three-dimensional hyperparameter space with one hyperparameter each for all tuning functions (**f**), all event-related and spike-history temporal filters (**g**, **h**), and all coupling filters (**p**). The value of the hyperparameters dictates the bias-variance trade-off in the model: whereas large values yield flat tuning functions and predict responses that lack task-specificity, small values will lead to poor test performance due to over-fitting. The optimal setting was found to be identical ([$\lambda_k$=100, $\gamma_l=\alpha$=10, $\beta$=10]) for the vast majority of neurons in the subset. Therefore, we used these values for fitting all neurons in the data as described below.

We fit several models by choosing different combinations of variables, performed 10-fold cross-validation to compute model likelihoods in each case, and selected the combination with the highest likelihood by the method of *Backward Elimination* which removed variables that did not contribute to improving the model likelihood. Because the model contained a large number coupling filters (equal to population size), these filters were selected as one group in the elimination process to minimize the computational complexity of the fitting procedure. Each fold of cross-validation comprised 9% of the trials, such that model selection was done using 90% of the data. The remaining 10% was used to evaluate the variance explained by the best-fit model. We estimated the variance explained (Pseudo-$R^2$) $\mathcal{M}$ as $\left[1 - \frac{(L_\infty - L_\mathcal{M})}{(L_\infty - L_0)}\right]$ where $L_\mathcal{M}$ is the log-likelihood of the model obtained by setting the mean of the Poisson spiking process $\mu_t^i = \hat{r}(t)$, $L_\infty$ is the log-likelihood of a model with $\mu_t^i = r(t)$, and $L_0$ is the log-likelihood of a model with constant firing rate $\mu_t^i = b^i$. Note that $L_\infty$ is the maximum possible log-likelihood achievable by any Poisson spiking model, while $L_0$ is the maximum possible log-likelihood achievable by a model with constant firing rate. The variance explained by any particular variable was estimated as the reduction in variance explained when that variable is removed from the model containing the set of all variables. We also estimated the fraction of variance explained in a more conventional way as the coefficient of determination ($R^2$) by comparing the raw firing rate (obtained by smoothing the observed spike train with a 60ms wide Gaussian) and model-estimated firing rates, and found qualitatively very similar results. We therefore used this latter measure, $R^2 = 1 - \frac{\mathrm{Var}(\hat{r}^i - r^i)}{\mathrm{Var}(r^i)}$, for reporting variance explained in single neurons throughout the text. Variance explained in the structure of population response was computed using an expression similar to coefficient of determination, except the numerator and denominator were both summed across neurons, $R^2_{\mathrm{pop}} = 1 - \frac{\sum_i \mathrm{Var}(\hat{r}^i - r^i)}{\sum_i \mathrm{Var}(r^i)}$. This measure is influenced more by the model's ability to explain responses of neurons with larger intrinsic variability. This is motivated by the fact that if most of the fluctuations in population activity is driven by a tiny fraction of neurons, then capturing the responses of those neurons is more critical to explaining the structure of population response.

**Recurrent neural network model.** We trained the RNN model defined in Eq. (3) by learning the recurrent weights $W^{\mathrm{rec}}$ using *BackPropagationThroughTime* (BPTT). On each trial, the 2D target location was encoded by the amplitudes of a 300 ms pulse arriving at two of the input channels (**x**). Target locations were drawn from the possible locations spanning the same range of distances and angles as monkey experiments, and varied randomly across trials. The network output (**y**) corresponded to the 2D hand velocity, such that an output of zero is akin to holding the joystick at a fixed position and would produce no change in the velocity of self-motion. Non-zero output, on the other hand, would result in a change in motion velocity. In this sense, the output of the network encoded acceleration and therefore integrated twice to compute the 2D position **s**. The network was trained to reach the target location (**x**) within a certain time $t^*$ and stay there for 0.6s (maximum stopping duration for monkeys). $t^*$ corresponded to the time taken when traveling along an idealized circular trajectory from the starting location to target location at maximum speed. To simulate sensory feedback in the form of optic flow, we integrated the network output once to compute 2D self-motion velocity, and fed it back to the remaining two input channels (**z**) with a small amount of sensory noise. The time-constant $\tau$ was set to 20 ms and each training trial lasted between 2-3 s depending on the target location. Weights were updated at the end of each trial by computing their gradients with respect to the loss function, $\mathcal{L} = \sum_k \sum_{t > t^*} |s_k(t) - x_k(t)|^2$, using BPTT. We also trained a variant of this model where the loss function contained additional terms that penalized high amplitudes and fast fluctuations in the network output and activity ($||\mathbf{y}||^2$, $||\dot{\mathbf{y}}||^2$, $||\mathbf{r}||^2$, $||\dot{\mathbf{r}}||^2$). This variant had smoother and sparser activity profiles, and exhibited sequential dynamics that was more comparable with the neural data. In all cases, small amount of process noise was added to the motor output channels (i.e., noisy plant) during training, to prevent the network from learning a purely autonomous control policy. Training was halted to probe the resulting neural representation once the performance reached the level of the average monkey.

The network was then retrained to be robust to task manipulations by fixing the input and output weights, and updating only the recurrent weights $W^{\mathrm{rec}}$. Sensory reliability was manipulated by increasing the amount of noise added to the sensory feedback channels (**z**). Motor gain was manipulated by multiplying the network output (**y**) by a gain factor before feeding it to the plant. To perturb the latent state dynamics, we added gaussian temporal pulses to the sensory feedback channels (**z**) at a random time after the target onset. Because adding sensory noise did not adversely affect performance, the network was tested on this manipulation without retraining. Since we do not know the precise change in signal-to-noise ratio that corresponds to density manipulation in the monkey experiments, we added the amount of noise that caused the performance level to fall off to the same extent as monkeys. For the remaining manipulations, the network was retrained until the performance reached the same level as the sensory manipulation condition.

## Statistical analysis
**Data exclusion.** Since we were interested in understanding latent state computation, we wanted to exclude data where the monkey was clearly not performing this computation. From each experimental recording session, we therefore excluded a small minority (~15%) of trials where the monkey appeared to clearly disengage from the task. Such trials were objectively identified as those in which the monkey either remained stationary throughout or failed to stop moving before the trial timed-out. This is analogous to the standard practice of excluding trials in which monkeys break fixation in more controlled experiments.

**Behavior.** In a co-ordinate system where the monkey's starting position was taken to be the origin, we evaluated behavioral performance by regressing each monkey's response positions ($r,\theta$) against target positions ($r^*,\theta^*$) separately for the radial ($r$ vs $r^*$) and angular ($\theta$ vs $\theta^*$) co-ordinates. The precision of the responses depended on the target location. To quantify the performance across all target

locations in a concise manner, we pooled all trials and performed ROC analysis as follows. For each session, we first constructed a psychometric function by calculating the proportion of correct trials as a function of (hypothetical) reward boundary which was varied between 0-4 m. Whereas an infinitesimally small boundary will result in all trials being classified as incorrect, a large enough reward boundary will yield near-perfect accuracy. To define a chance-level psychometric function, we repeated the above procedure but now by shuffling the target locations across trials, thereby destroying the relationship between target and response locations. Finally, we obtained the ROC curve by plotting the proportion of correct trials in the original dataset (true positives) against the shuffled dataset (false positives) for each value of hypothetical reward boundary. We used the area under this ROC curve to obtain an accuracy measure as a single scalar value for each recording session.

**Neural sequences.** Peak-normalized response of neurons were first calculated by averaging responses by grouping trials according to target distance (nearby vs distant) and target angle (leftward vs rightward). Spike times were re-scaled based on the trial duration before trial-averaging, and the response profile of each neuron was subsequently normalized by the peak activity. Neurons were sorted according to the timing of their peak response observed within each trial group to construct firing rate maps of sequential activity. Pattern similarity was defined as the correlation coefficient between the firing rate maps taken from either the same trial group (odd vs even trials) or different trial groups (nearby vs distant targets, or leftward vs rightward targets). Time course of the pattern similarity was computed as the correlation between population activity vectors (columns of the rate maps) taken from the same trial group (odd vs even trials) or different trial groups (nearby vs distant targets, or leftward vs rightward targets). Following ref. [74], Sequentiality index (Sql) was defined as the geometric mean of peak sparseness ($f_{peak}$) and temporal sparseness ($f_{temp}$), $\text{Sql} = \sqrt{f_{temp} * f_{peak}}$ where:

$$f_{peak} = \sum_{t=1}^{M} -p_t \log(p_t) / \log(M) \tag{6.1}$$

$$f_{temp} = 1 - \mathbb{E}_t \left[ \sum_{i=1}^{N} -r_i^t \log(r_i^t) / \log(N) \right] \tag{6.2}$$

where $M$ and $N$ denote the number of time bins and neurons respectively, $p_t$ is the fraction of neurons whose activity peaked in time bin $t$, $r_i^t$ denotes the activity of neuron $i$ in time bin $t$, normalized by the sum of activities of all neurons in that bin. Peak sparseness is high if the distribution of the time of peak activity across the population is roughly uniform. Temporal sparseness is high if only a few neurons are active in each time bin.

**Cross-correlation function.** The cross-correlation between spike trains $r_i(t)$ and $r_j(t)$ was computed as $R_{ij}(\tau) = \frac{1}{N\bar{r}_i}(\sum_t r_i(t) r_j(t-\tau)) - \bar{r}_i$ where $\bar{r}_i$ denotes the time-averaged firing rate of neuron $i$. This can be interpreted as the excess spike rate in neuron $i$ due to neuron $j$[36]. The auto-correlation function was a special case corresponding to $i=j$.

**Stability index.** The stability of coupling filters and tuning functions to task manipulations was quantified using Stability index (SI). Stability index was of coupling filters was given by $\frac{\rho - \rho_0}{\rho^* - \rho_0}$ where $\rho$ is the median correlation between coupling filters fit to data with and without task manipulation, $\rho^*$ is the median correlation between coupling filters fit to data in odd and even trials of the baseline task, and $\rho_0$ is the median of the null distribution constructed by shuffling neuronal pairs. A value of 0 corresponds to highly unstable coupling where the degree of

match to baseline condition is no better than chance, and 1 corresponds to perfectly stable coupling where the filter did not change shape. SI of tuning functions was determined in an analogous manner where $\rho$ denoted the correlation between tuning functions of a neuron in data with or without task manipulations. Note that depending on the type of task manipulation, the distribution of some of the task variables would change a lot. For example, the sensory input (velocity) is scaled by 2x during gain manipulation. To keep things consistent across all analyses, we fixed the domain over which tuning functions were computed to be identical to the domain used when fitting the model using baseline data.

**Mixed selectivity index.** Mixed selectivity index was used to estimate the uniformity of variance explained by different task variables in the neuronal response (Fig. 7). It was quantified as the participation ratio, $[\sum_{i=1}^{K} v_i]^2 / \sum_{i=1}^{K} [v_i]^2$ where $v_i$ denotes the variance explained by task variable $i$, and $K = 6$ is the number of task variables. This index is bounded between 1 (no mixing where only one variable contributes to predicting neural activity) and $K$ (uniform mixing where all variables contribute equally to the predicting neural activity).

**Canonical correlation analysis.** We used canonical correlation analysis (CCA), an iterative technique to assess task-relevant linear dimensionality of population response (Fig. S4d). We considered the set of all continuous-valued variables except for LFP phase resulting in a total of $N = 6$ task-relevant variables. We considered the set of all simultaneously recorded neurons resulting in an $M$ dimensional vector of neural activity at each time step ($M \gg N$). If $X$ and $R$ denote the time-course of the set of all task variables and population response respectively, we first identify a pair of vectors $a \in \mathbb{R}^N$ and $b \in \mathbb{R}^M$ that maximizes the correlation, $\text{Corr}(a^\mathsf{T} X, b^\mathsf{T} R)$, between the pair of canonical variables obtained by projecting the task and neural response variables onto the directions specified by those vectors. Then, we identify a second pair of vectors in the same way but with the additional constraint that the resulting canonical variables are uncorrelated with the first pair of canonical variables. We continue this procedure $N$ times to identify up to $N$ task-relevant dimensions of neural response. Dimensionality of canonical correlations is classically defined simply as the number of canonical pairs with significant correlations. However, this measure of dimensionality fails to account for the differences between the actual fraction of covariance in those dimensions. To capture the spectrum of covariance between task variables and neural response, we instead defined "task-relevant neural dimensionality" analogously to the standard measure of participation ratio used to measure the flatness of eigenspectra.

$$D = \frac{\left[ \sum_{i=1}^{P} \text{Cov}(a_i^\mathsf{T} X, b_i^\mathsf{T} R) \right]^2}{\sum_{i=1}^{P} \left[ \text{Cov}(a_i^\mathsf{T} X, b_i^\mathsf{T} R) \right]^2} \tag{6.3}$$

where $a_i$ & $b_i$ correspond to the $i$th canonical pair of vectors with unit-norm, and $M$ & $N$ denote the number of task variables and neurons, $P = \min(M, N)$, and $1 \le D \le P$.

### Reporting summary
Further information on research design is available in the Nature Portfolio Reporting Summary linked to this article.

## Data availability
Pre-processed data are available at https://gin.g-node.org/kaushik-l/firefly-monkey. Raw data (~50 GB per experimental session) are stored in a local database server but available upon reasonable request. Source data files are provided with this paper. Source data are provided with this paper.

## Code availability

The codes used to perform the analyses in this study are available at https://github.com/kaushik-l/neuroGAM, https://github.com/kaushik-l/firefly-monkey, and https://github.com/DeepLabCut/DeepLabCut/releases/tag/1.11.

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

## Acknowledgements

The authors express their deepest gratitude to Roozbeh Kiani for performing the Utah array implantations, Erin Neyhart for assisting with the data collection, Karen Wood and Rebecca Meyer for assisting with spike-sorting, and Jing Lin and Jian Chen for their help in programming the stimulus. This work was supported by NIH grant 1R01 DC004260 and 1R01 NS127122 to D.E.A., NSF NeuroNex 1707400 and NSF CAREER IOS-1552868 to X.P, NIH CRCNS 1R01 NS120407-01, 1U19 NS118246, and Simons Collaboration on the Global Brain, grant no. 324143 to X.P. and D.E.A. K.J.L. was supported by the NSF NeuroNex Award DBI-1707398 and the Gatsby Charitable Foundation GAT3780.

## Author contributions

K.J.L., X.P., and D.E.A. conceived the study; K.J.L. and E.A. adapted the paradigm to non-human primates; E.A. developed the protocol to train monkeys to perform dexterous joystick movements and participated in the surgeries for implanting multi-electrode arrays; K.J.L. and E.A. collected the data with the help of research technicians. E.A. performed spike-sorting with the help of research technicians. K.J.L. performed the data analysis and wrote the manuscript. K.J.L., E.A., X.P., and D.E.A. revised the manuscript.

## Competing interests

The authors declare no competing interests.
