## [Peer Review File · Nature Communications]

Dynamical Latent State Computation in the Male Macaque Posterior Parietal CortexREVIEWER COMMENTS

Reviewer #1 (Remarks to the Author):

This paper uses a new behavioral task to study the representation of latent states in the primate posterior parietal cortex. The authors use a series of behavioral analyses, neural recordings, encoding and decoding analyses, and RNN modeling. I enjoyed reading this paper and feel it adds important and interesting results to the field. In particular, the behavioral task is a clever design and a new approach for studying PPC in monkeys. The analysis approaches are well matched to the recordings and a breath of fresh air for the field of monkey PPC research. The encoding analyses provide an overview of the entirety of the neural activity instead of focusing solely on a targeted sliver of the data. Also, the decoding approaches during overshoot/undershoot trials and experimental manipulations are clever ways to refine the interpretation of the PPC spiking. The results and study design help to bridge between previous work in PPC in rodents and monkeys, and I found it interesting to see spiking and encoding in the monkey PPC that is reminiscent of reports in rodent PPC. I am excited about the direction of this work and hope it opens the doors for others in the field to follow this type of experimental and analysis design. The text and figures are presented clearly, and the results and interpretations are well supported by the data.

I do not have any specific concerns that need to be addressed prior to publication. The questions that came to mind are all extensions of the analyses presented. There is more that can be done with the neural and behavioral data. However, these questions all extend beyond, and do not affect, the message of the paper presented here. In the spirit of focusing peer review on addressing the validity of the claims presented, I therefore decided to reserve these questions. I support publication of this paper and congratulate the authors on an excellent manuscript.

Reviewer #2 (Remarks to the Author):

Summary

The authors aim to understand how the posterior parietal cortex in macaque monkeys can support computations that encode the state of the world, even when the state must be updated or inferred based on indirect stimulus information. To accomplish this, they used a continuously valued navigation task. On each trial, the monkey was asked to steer towards a specified location in a 3D virtual environment using a joystick. However, the location was only displayed at the start of the trial and the

environment contained no fixed landmarks. As a result, the monkey had to remember and track the target location by integrating optic flow information. Populations of neurons were recorded during the task from arrays implanted in area 7a in the parietal cortex.

The analysis of the behavior in this task is presented very thoroughly, which provides a helpful demonstration of how to analyze more complex tasks with continuously valued responses. Statistical analyses of neural populations recorded during the task using both generalized additive models and decoding methods were consistent with the hypothesis that the posterior parietal cortex tracks the location of the (hidden/latent) target location using optic flow. Additionally, recurrent neural networks trained to perform the same task showed consistent deficits in performance as the monkeys after manipulating aspects of the task (stimulus noise, motor output gain).

All together, this study provides a valuable advance for studying sensorimotor computations and the function of the posterior parietal cortex beyond the existing literature. While I believe this paper will ultimately be a good contribution to Nature Communications, I have a few technical points to address before it is ready for publication.

Major Concerns

The introduction did not clearly motivate why the authors recorded from 7a in particular. The references on line 36 are appropriate, but more details about why 7a is a good target for this study could be fleshed out here in the introduction, including why 7a was chosen over other parietal areas.

There are a number of technical details concerning the model fitting and evaluation with the GAM that should be clarified and checked:

Were the reported pseudo- R^2 values reported fully cross-validated? The methods section (line 622) says cross-validation was performed for model selection. It's unclear if this training R^2 was what was reported or not. Either a withheld test set or multi-level cross-validation (as was described for the decoder analysis, lines 583-595) needs to be used.

Were the train/test splits for cross-validation taken across trials (like was done for decoding)? This is because strong correlations over time within trials may inflate the actual predictiveness of coupling by effectively interpolating with many parameters (and may result in longer timescale filters).

How were the regularization hyperparameters selected (line 608)?

Were coupling filters individually selected for or were they selected as a group in the backward elimination procedure?

How were the tuning functions (f,g) parameterized? With a basis set? I only noticed the definition of the coupling basis (in the figure 4E caption, not the methods).

The tuning curve function in equation 5 is incorrect. The right side as written is actually $\exp(E[\log(\mu)])$, which by Jensen's inequality is less than or equal to $E[\mu]$ (the left side).

The spike history and coupling effects are ignored in this equation. Is this computed from the model fit without those?

Either way, some clarification would help with completeness.

Line 145 and supplementary fig S3B middle refer to the percent "variance in the structure of population response" rather than single neurons. How is the quantity computed?

Decoder performance in 6B as a correlation coefficient: this plot reads as if the correlation coefficient was fit to each trial and averaged. Such a procedure could inflate performance (by effectively fitting 2 more parameters for each trial). Is this the case? Why not the coefficient of determination (R^2) instead?

With the RNN manipulations, is adding noise to the stimulus comparable to the optic flow manipulation the monkeys experienced? Another possibility is that this could act like a gain (instead or in addition to the additive noise). A gain change on the stimulus may result in different behavior in the RNN.

Minor Points

I appreciate that the authors make the distinction between GLMs and GAMs (which is typically ignored). However, to be clearer to readers, the authors should point out the connection to GLMs so that the readers know this isn't something totally different than existing GLM+coupling analyses that they might be familiar with. Some spike train GLMs are formulated as GAMs: for example, a "linear filter" for a delta function event (for example, see fig 2B in Park et al 2014 [ref 65], top and bottom events in the left column) could really be given as a function over time in a GAM.

Line 170 - are the means and SDs of the likelihoods across neurons, trials?

The Figure 5A caption refers to the decoder as “generalized least squares”, but the methods section shows the OLS estimator on line 591. Is it proper to call this generalized least squares here? Could the justification for this choice be that the Gaussian smoothing width on the rates is a hyperparameter?

One suggestion: the use of the word “latent” for a coefficient in the model was often confusing: in the context of the GAM, I kept having to tell myself that it was a fixed coefficient and not a latent variable in the statistical sense to be marginalized out or estimated. It makes sense in the intro talking about the task and motivation, but for the statistical modeling portion it may be clearer to use a different term like “displacement”.

The GAM analysis refers to “excitatory” and “inhibitory” coupling. However, the coupling in the model is statistical, and doesn’t necessarily correspond to excitatory and inhibitory synapses. This distinction could be pointed out when the results are presented. or an alternative term like “suppressive” could be used to distinguish the statistical effect from the physiological term.

REVIEWER COMMENTS

Reviewer #1 (Remarks to the Author):

1. This paper uses a new behavioral task to study the representation of latent states in the primate posterior parietal cortex. The authors use a series of behavioral analyses, neural recordings, encoding and decoding analyses, and RNN modeling. I enjoyed reading this paper and feel it adds important and interesting results to the field. In particular, the behavioral task is a clever design and a new approach for studying PPC in monkeys. The analysis approaches are well matched to the recordings and a breath of fresh air for the field of monkey PPC research. The encoding analyses provide an overview of the entirety of the neural activity instead of focusing solely on a targeted sliver of the data. Also, the decoding approaches during overshoot/undershoot trials and experimental manipulations are clever ways to refine the interpretation of the PPC spiking. The results and study design help to bridge between previous work in PPC in rodents and monkeys, and I found it interesting to see spiking and encoding in the monkey PPC that is reminiscent of reports in rodent PPC. I am excited about the direction of this work and hope it opens the doors for others in the field to follow this type of experimental and analysis design. The text and figures are presented clearly, and the results and interpretations are well supported by the data.

We thank the reviewer for recognizing the novelty and significance of this paradigm. Bridging the gap between rodent and primate studies of PPC was indeed one of the goals of this study, a fact that we have now highlighted in the discussion section (lines 381-382) of the revised manuscript.

2. I do not have any specific concerns that need to be addressed prior to publication. The questions that came to mind are all extensions of the analyses presented. There is more that can be done with the neural and behavioral data. However, these questions all extend beyond, and do not affect, the message of the paper presented here. In the spirit of focusing peer review on addressing the validity of the claims presented, I therefore decided to reserve these questions. I support publication of this paper and congratulate the authors on an excellent manuscript.

Thanks for the encouragement to build on the analysis presented here to gain more insights into the behavioral strategy and the role of PPC. We are enthusiastic about sharing these insights in future work.

Reviewer #2 (Remarks to the Author):

Summary

The authors aim to understand how the posterior parietal cortex in macaque monkeys can support computations that encode the state of the world, even when the state must be updated or inferred based on indirect stimulus information. To accomplish this, they used a continuously valued navigation task. On each trial, the monkey was asked to steer towards a specified location in a 3D virtual environment using a joystick. However, the location was only displayed at the start of the trial and the environment contained no fixed landmarks. As a result, the monkey had to remember and track the

target location by integrating optic flow information. Populations of neurons were recorded during the task from arrays implanted in area 7a in the parietal cortex.

The analysis of the behavior in this task is presented very thoroughly, which provides a helpful demonstration of how to analyze more complex tasks with continuously valued responses. Statistical analyses of neural populations recorded during the task using both generalized additive models and decoding methods were consistent with the hypothesis that the posterior parietal cortex tracks the location of the (hidden/latent) target location using optic flow. Additionally, recurrent neural networks trained to perform the same task showed consistent deficits in performance as the monkeys after manipulating aspects of the task (stimulus noise, motor output gain).

All together, this study provides a valuable advance for studying sensorimotor computations and the function of the posterior parietal cortex beyond the existing literature. While I believe this paper will ultimately be a good contribution to Nature Communications, I have a few technical points to address before it is ready for publication.

We thank the reviewer for their positive assessment and accurate summary of our work. We have carefully considered all concerns and addressed them to the best of our ability. Below, we include point-by-point response along with the associated changes to the manuscript file, wherever applicable.

Major Concerns

1. The introduction did not clearly motivate why the authors recorded from 7a in particular. The references on line 36 are appropriate, but more details about why 7a is a good target for this study could be fleshed out here in the introduction, including why 7a was chosen over other parietal areas.

We have added a paragraph in the introduction (lines 37-53) of the revised manuscript to provide a detailed explanation for why area 7a was chosen over other areas of parietal cortex. Our goal was to study neural computations that contribute to tracking the latent world state (position) by integrating movement velocity. However, to do so, monkeys must first estimate velocity from the pattern of optic flow. Because the mechanism of optic flow processing was not the subject of our investigation, we wanted to choose a brain area that likely already receives abstract velocity signals. There were several properties that made area 7a a suitable choice. First, anatomical tracing studies have consistently found a pattern of inter-areal connectivity that puts area 7a at the top of the motion-processing ('dorsal stream') hierarchy. Moreover, it is one of the few areas in PPC that directly projects to the hippocampal formation, with lesions to area 7a affecting navigation performance. Second, area 7a neurons are known to have large, bilateral receptive fields (spanning 30-50 degrees) that are activated by the full-field motion stimuli such as the one used in this study. Third, response properties of area 7a neurons indicate that they are capable of marginalizing away the influence of eye movements thereby representing visual inputs in a navigationally useful (non-retinotopic) format. Fourth, we confirmed in prior work that neurons in area 7a indeed encode linear and angular velocity in an abstract format, regardless of stimulus modality, under passive viewing conditions. Finally, previous work has shown that representation of cognitive variables in area 7a is clearly decoupled from the influence of sensory and

motor variables whereas such decoupling has not been demonstrated elsewhere. Although some of the above properties might be shared with other brain areas, evidence available so far pointed to area 7a as the main candidate for latent state computation in our experimental paradigm.

There are a number of technical details concerning the model fitting and evaluation with the GAM that should be clarified and checked:

We thank the reviewer for providing an opportunity to add several missing details to the methods section and to correct a technical error.

2. Were the reported pseudo- R^2 values reported fully cross-validated? The methods section (line 622) says cross-validation was performed for model selection. It's unclear if this training R^2 was what was reported or not. Either a withheld test set or multi-level cross-validation (as was described for the decoder analysis, lines 583-595) needs to be used.

The R^2 values reported in line 148 of the original manuscript correspond to the average R^2 across test sets (or more precisely, validation sets) from different folds of the cross-validation. But since these were the same test sets used for likelihood-based model selection, the reported R^2 values were not indicative of the generalization performance of the selected models.

In the updated manuscript, we refit the model by setting aside 10% of the data as a separate test set, and report R^2 values estimated on this withheld test set. These fully cross-validated R^2 values (reported in line 163 of the revised manuscript) are slightly lower than the values reported previously, but the tuning statistics (reported later in the same section) remained unchanged because the model selection procedure was robust to withholding 10% of the data.

3. Were the train/test splits for cross-validation taken across trials (like was done for decoding)? This is because strong correlations over time within trials may inflate the actual predictiveness of coupling by effectively interpolating with many parameters (and may result in longer timescale filters).

Yes, the train/test splits were taken across trials in a similar manner for encoding and decoding models. This is clarified in lines 669-671 of the revised manuscript. The only difference in the fitting procedure between encoding and decoding models was that encoding model was fit using 10-fold cross validation whereas decoding model was fit using a single validation set.

4. How were the regularization hyperparameters selected (line 608)?

We apologize for not providing this detail. Hyperparameters for regularization were determined using a separate cross validation procedure on a subset of neurons, prior to backward elimination. In this procedure, we varied the hyperparameter values on a logarithmic scale from 0.001 to 1000 and fit the model by including all task variables for each hyperparameter setting using 90% of the data. The hyperparameter combination with the highest model likelihood in the remaining 10% of the data was chosen as the optimal setting. To reduce the complexity of this procedure, we assumed a three-

dimensional hyperparameter space with one hyperparameter each for all tuning functions (**f**), all event-related and spike-history temporal filters (**g**, **h**), and all coupling filters (**p**). The optimal setting was found to be identical ([100, 10, 10]) for most neurons in the subset. Therefore, we used these values for the backward elimination procedure. This is described in the Methods line 655-665 of the revised manuscript.

5. Were coupling filters individually selected for or were they selected as a group in the backward elimination procedure?

Coupling filters were selected as a group in order to keep the computation time manageable. All other variables were individually selected for. This is now stated in the methods section line 669-670 of the revised manuscript.

6. How were the tuning functions (**f**,**g**) parameterized? With a basis set? I only noticed the definition of the coupling basis (in the figure 4E caption, not the methods).

We apologize for the omitting this important detail. The updated Methods section includes a detailed description of the parameterization used for different types of predictors in lines 647-654 of the revised manuscript. Briefly, tuning functions were parameterized by binning the feature space and using a basis of one-hot vectors to encode different bins. The event-related temporal filters were parameterized using a raised cosine basis. Coupling and spike-history filters were parameterized using a raised cosine basis on a logarithmic scale.

7. The tuning curve function in equation 5 is incorrect. The right side as written is actually $\exp(E[\log(\mu)])$, which by Jensen's inequality is less than or equal to $E[\mu]$ (the left side).

The reviewer is correct that the tuning functions plotted in figure 3 corresponded only to a lower-bound, and we thank them for catching this mistake. In the updated manuscript (equation 5), we recalculate the tuning functions by taking $E[\mu]$. This correction was found to amplify the tuning functions by a small factor, which affects the example responses plotted in Figure 3F, but not the statistics reported in the GAM section main text such as variance explained and the fraction of tuned neurons. The stability index of tuning functions (lines 284-285) have also been modified to reflect this although the change is negligible.

8. The spike history and coupling effects are ignored in this equation. Is this computed from the model fit without those? Either way, some clarification would help with completeness.

Yes, the reviewer is correct. The expectations were computed by ignoring the effects of spike-history and coupling filters as the modulatory influence these filters had on the tuning function was close to 1. This is clarified in the methods lines 634-636 of the revised manuscript.

9. Line 145 and supplementary fig S3B middle refer to the percent “variance in the structure of population response” rather than single neurons. How is the quantity computed?

The coefficient of determination for single neurons is defined as (one minus) the ratio between mean squared estimation error across time and the variance of the neuronal activity across time. Variance explained in the structure of population response was computed using an expression similar to coefficient of determination, except the numerator and denominator were both summed across neurons, such that this measure is influenced more by the model's ability to explain responses of neurons with larger intrinsic variability. This definition is motivated by the fact that if most of the fluctuations in population activity is driven by a tiny fraction of neurons, then capturing the responses of those neurons is more critical to explaining the structure of population response. This definition is now provided in methods lines 681-686 of the revised manuscript.

10. Decoder performance in 6B as a correlation coefficient: this plot reads as if the correlation coefficient was fit to each trial and averaged. Such a procedure could inflate performance (by effectively fitting 2 more parameters for each trial). Is this the case? Why not the coefficient of determination (R^2) instead?

The correlation coefficient was computed only after concatenating all the trials, so the estimates are not confounded by the problem identified by the reviewer. We apologize for not making this clear. We used correlation coefficient as it can be readily interpreted as the degree of alignment between the decoded estimates and the ground truth. The coefficient of determination (R^2) suggested by the reviewer turns out to be equal to the square of the correlation coefficient reported in the manuscript.

11. With the RNN manipulations, is adding noise to the stimulus comparable to the optic flow manipulation the monkeys experienced? Another possibility is that this could act like a gain (instead or in addition to the additive noise). A gain change on the stimulus may result in different behavior in the RNN.

In previous work, we showed using computational modeling that behavioral consequence of optic flow manipulation in humans can be well explained by changing the width of the stimulus likelihood function (Lakshminarasimhan et al. 2018). This is equivalent to changing the variance of the observation noise. Therefore, we simulated optic flow manipulation by increasing the variance of noise added to velocity input channel. A second manipulation where we increase the sensitivity of the joystick (for monkeys) was simulated by changing the multiplicative gain (for RNN) and indeed produced a qualitatively different behavior from optic flow manipulation in both monkeys and in the RNN. Specifically, whereas optic flow manipulation in monkeys (and increasing additive noise to RNN input) increased the variability in stopping position without affecting travel duration, manipulating joystick sensitivity (and multiplicative gain to RNN input) altered the average travel duration as shown in the figure below.

The reviewer might be correct that additive noise to RNN could alter the response gain of the RNN units. However, since the behavior of the RNN models is similar to monkeys, we believe they offer useful insights into neural mechanisms.

Minor Points

12. I appreciate that the authors make the distinction between GLMs and GAMs (which is typically ignored). However, to be clearer to readers, the authors should point out the connection to GLMs so that the readers know this isn't something totally different than existing GLM+coupling analyses that they might be familiar with. Some spike train GLMs are formulated as GAMs: for example, a "linear filter" for a delta function event (for example, see fig 2B in Park et al 2014 [ref 65], top and bottom events in the left column) could really be given as a function over time in a GAM.

We thank the reviewer for this suggestion. For the benefit of readers already more familiar with GLMs, we have added a sentence at the end of the paragraph (line 159-160 of the revised manuscript) to mention the relationship to GLM.

13. Line 170 - are the means and SDs of the likelihoods across neurons, trials?

Mean and SD are both across neurons. This is now stated in line 188 of the revised manuscript.

14. The Figure 5A caption refers to the decoder as "generalized least squares", but the methods section shows the OLS estimator on line 591 Is it proper to call this generalized least squares here? Could the justification for this choice be that the Gaussian smoothing width on the rates is a hyperparameter?

We apologize for the typo. The decoder was determined using OLS as the reviewer correctly points out. In earlier iterations of this analysis, we also fit alternative models including generalized least squares that took the covariance of the residuals into account. Since we found qualitatively similar results to OLS, we chose to report the results of OLS as it simpler and more widely used. We have corrected the typo in the caption of Figure 5A in the revised manuscript.

15. One suggestion: the use of the word "latent" for a coefficient in the model was often confusing: in the context of the GAM, I kept having to tell myself that it was a fixed coefficient and not a latent variable in the statistical sense to be marginalized out or estimated. It makes sense in the intro talking about the task and motivation, but for the statistical modeling portion it may be clearer to use a different term like "displacement".

We thank the reviewer for this important feedback on potentially confusing language. We have now limited the use of the word "latent", to the first paragraph of the GAM section where we introduce different predictors.

16. The GAM analysis refers to "excitatory" and "inhibitory" coupling. However, the coupling in the model is statistical, and doesn't necessarily correspond to excitatory and inhibitory synapses. This

distinction could be pointed out when the results are presented. or an alternative term like “suppressive” could be used to distinguish the statistical effect from the physiological term.

The reviewer is correct that the valence of coupling is purely statistical. However, since the use of excitatory/inhibitory classification scheme in this statistical model is motivated by biology, we feel it is appropriate to use these terms. To ensure that our terminology is not misconstrued by the reader, we have added a sentence in that section to point out that these filters are not meant to capture synaptic properties (line 202-204 of the revised manuscript).

REFERENCES:

Lakshminarasimhan KJ, Petsalis M, Park H, DeAngelis GC, Pitkow X, Angelaki DE. A Dynamic Bayesian Observer Model Reveals Origins of Bias in Visual Path Integration. *Neuron*, 99(1):194–206.e5, 2018

REVIEWERS' COMMENTS

Reviewer #3 (Remarks to the Author):

The authors have thoroughly addressed the technical comments given in the first round, and I am happy to give a very brief review. I strongly recommend the revised manuscript for publication in Nature Communications as is. The question addressed by the study - neural dynamics supporting internal tracking of behaviorally relevant variables - remains a good fit for this journal. The study was well designed, motivated, and presented.

REVIEWERS' COMMENTS

Reviewer #3 (Remarks to the Author):

The authors have thoroughly addressed the technical comments given in the first round, and I am happy to give a very brief review. I strongly recommend the revised manuscript for publication in Nature Communications as is. The question addressed by the study - neural dynamics supporting internal tracking of behaviorally relevant variables - remains a good fit for this journal. The study was well designed, motivated, and presented.

- We thank the reviewer for their positive assessment and for their valuable feedback in the first round.